



# SiDroForest: A comprehensive forest inventory of Siberian boreal forest investigations including drone-based point clouds, individually labelled trees, synthetically generated tree crowns and Sentinel-2 labelled image patches

5 Femke van Geffen[1,2], Birgit Heim[1], Frederic Brieger[1,3], Rongwei Geng[1,4,5], Iuliia A. Shevtsova[1,2], Luise Schulte[1,2], Simone M. Stuenzi[1,6], Nadine Bernhardt[1,7], Elena I. Troeva[8], Luidmila A. Pestryakova[9], Evgenij S. Zakharov[8,9], Bringfried Pflug[10], Ulrike Herzschuh[1,2,11], Stefan Kruse[1]

1 Alfred Wegener Institute Helmholtz Centre for Polar and Marine Research (AWI), Research Unit   Potsdam, Germany

10 2 University of Potsdam, Institute of Biochemistry and Biology, Potsdam, Germany

3 Carleton University, Department of Geography and Environmental Studies Ottawa, Canada

4 Key Laboratory of Land Surface Pattern and Simulation, Institute of Geographical Sciences and Natural Resources Research, Chinese Academy of Sciences, Beijing, China

5 University of Chinese Academy of Sciences, Beijing, China

15 6 Humboldt-Universität zu Berlin, Geography Department, Unter den Linden, Berlin, Germany

7 Julius Kühn-Institut Bundesforschungsinstitut für Kulturpflanzen, Quedlinburg, Germany

8 Institute for Biological Problems of the Cryolithozone, Russian Academy of Sciences, Siberian Branch, Yakutsk, Russia

9 North-Eastern Federal University of Yakutsk, Institute of Natural Sciences, Yakutsk, Russia

20 10 German Aerospace Center (DLR), Berlin, Germany

University of Potsdam, Institute of Environmental Science and Geography, Potsdam, Germany

*Correspondence to*: Stefan Kruse (stefan.kruse@awi.de), Femke van Geffen (femke.van.geffen@awi.de)





## Abstract

This data collection is an attempt to remedy the scarcity of tree level forest structure data in the circum-boreal region, whilst providing, as part of the data collection, adjusted and labelled tree level and vegetation plot level data for machine learning and upscaling practices. Publicly available comprehensive datasets on tree level forest structure are rare, due to the involvement of governmental agencies, public sectors, and private actors that all influence the availability of these datasets.

We present datasets of vegetation composition and tree and plot level forest structure for two important vegetation transition
zones in Siberia, Russia; the summergreen–evergreen transition zone in central Yakutia and the tundra–taiga transition zone in Chukotka (NE Siberia). The SiDroForest collection contains a variety of data mainly based on unmanned aerial vehicle (UAV) and field data collected from 64 vegetation plots during fieldwork jointly performed by the Alfred Wegener Institute for Polar and Marine Research (AWI) and the North-Eastern Federal University of Yakutsk (NEFU) during the Chukotka 2018 expedition to Siberia.

The data collection consists of four separate datasets. The fieldwork locations are the anchors that bind the data types together based on the location of the vegetation plot.

i) The first dataset (Kruse et al., 2021, https://doi.pangaea.de/10.1594/PANGAEA.933263) provides UAV-borne data products covering the 64 vegetation plots surveyed during fieldwork: including structure from motion (SfM) point clouds, point-cloud products such as Digital Elevation Model (DEM), Canopy Height Model (CHM), Digital Surface Model (DSM) and Digital
Terrain Model (DTM) constructed from Red Green Blue (RGB) and Red Green Near Infrared (RGN) orthomosaics. Forest structure and vegetation composition data are crucial in the assessment of whether a forest is to act as a carbon sink under changing climate conditions. Fieldwork and UAV-products can provide such data in depth.

ii) The second dataset contains spatial data in the form of points and polygon shape files of 872 labelled individual trees and shrubs that were recorded during fieldwork at the same vegetation plots with information on tree height, crown diameter, and
species (van Geffen et al., 2021c, https://doi.pangaea.de/10.1594/PANGAEA.932821). These tree- and shrub-individual labelled point and polygon shape files were generated and are located on the UAV RGB orthoimages. The individual number links to the information collected during the expedition such as tree height, crown diameter and vitality provided in table format. This dataset can be used to link individual trees in the SfM point clouds, providing unique insights into the vegetation composition and also allows future monitoring of the individual trees and the contents of the recorded vegetation plots at large.

iii) The third dataset contains a synthesis of 10 000 generated images and masks that have the tree crowns of two species of larch (*Larix gmelinii* and *Larix cajanderi*) automatically extracted from the RGB UAV images in the common objects in context (COCO) format (van Geffen et al., 2021a, https://doi.pangaea.de/10.1594/PANGAEA.932795). The synthetic dataset was created specifically to detect Siberian larch species.

iv) If publicly available forest-structure datasets at tree level are rarely available for Siberia, even fewer ready-to-use tree and
plot level data are available for machine learning approaches, for example optimised data formats containing annotated vegetation categories. The fourth set contains Sentinel-2 Level-2 bottom of atmosphere labelled image patches with seasonal





information and annotated vegetation categories covering the vegetation plots (van Geffen et al., 2021b, https://doi.pangaea.de/10.1594/PANGAEA.933268). The dataset is created with the aim of providing a small ready-to use validation and training data set to be used in various vegetation-related machine-learning tasks.

The SidroForest data collection serves a variety of user communities. First of all, the UAV-derived top of canopy structure information, orthomosaics and the detailed vegetation information in the labelled data set provide detailed information on forest type, structure and composition for scientific communities with ecological and biological applications. The detailed Land Cover and Vegetation structure information in the first two data sets are of use for the generation and validation of Land Cover remote sensing products in radar and optical remote sensing. In addition to providing information on forest structure and vegetation composition of the vegetation plots, parts of the SiDroForest dataset are prepared to be used as training and validation data for machine learning purposes. For example, the Synthetic tree crown dataset is generated from the raw UAV images and optimized to be used in neural networks. Furthermore, the fourth SiDroForest data set contains standardized Sentinel-2 labelled image patches that provide training data on vegetation class categories for machine learning classification with JSON labels provided. The SiDroForst data collective serves as a basis to add future data collected during expeditions performed by the Alfred Wegener Institute, creating a larger dataset in the upcoming years that can provide unique insights into remote hard to reach boreal regions of Siberia.

## 1 Introduction

Circumpolar boreal forests represent close to 30% of all forested areas and are changing in response to climate, with potentially important feedback mechanisms to regional and global climate through altered carbon cycles and albedo dynamics (e.g., Loranty et al., 2018). These forests are located primarily in Alaska, Canada, and Russia. Forests are three-dimensional systems whose biophysical structure plays major roles in ecosystem function and diversity. Forest structure is a crucial component in the assessment of whether a forest is likely to act as a carbon sink under changing climate (e.g., Schepaschenko et al., 2021). On one hand, publicly available comprehensive datasets on forest structure are rare, due to the involvement of governmental agencies, public sectors, and private actors who all influence the availability of these datasets. On the other hand, there are national and international collaborative initiatives that aim to distribute high quality forest data collections. That said, the Arctic-Boreal Vulnerability Experiment (ABoVE) run by the NASA Terrestrial Ecology Program provides open-source data collections from boreal and arctic regions in Alaska and Canada (ABoVE Science Definition Team, 2014). Also, the Forest Observation System (FOS, http://forest-observation-system.net/) that is supporting Earth Observation validation and algorithm development efforts such as described in Chave et al. (2019) provides a publicly available global Above Ground Biomass (AGB) database (Schepaschenko et al. 2019) containing a high number of plot level data from the boreal forest domain. Schepaschenko et al. (2017) used inventories from the old Soviet Forest Inventory and Planning System (FIP) and the new National Forest Inventory (NFI) to compile and publish a highly comprehensive forest AGB data collection at plot level specifically for Eurasia. This data collection (Schepaschenko et al., 2017) and FOS (Schepaschenko et al. 2019) both distribute



aggregated plot level. The data collection presented here provides open-source forest structure-related data at plot and also at tree level for boreal forests in central and North Eastern Siberia, Russia.

Still, few data are publicly available at tree level or plot level that are ready-to use for machine learning applications in the field of remote sensing, for example optimised data containing annotated vegetation categories. We therefore aim to provide tree level as well as plot level boreal forest datasets, with parts of the data collection that can be further used for remote-sensing applications and machine-learning analyses.

The data presented here includes a variety of data types contained in four datasets. The data collection is underpinned by fieldwork data from vegetation plots and unmanned aerial vehicle (UAV) acquisitions from a German-Russian expedition in 2018 by the Alfred Wegener Institute for Polar and Marine Research (AWI) together with the North-Eastern Federal University of Yakutsk (NEFU) to central and eastern Siberia. Likethis, the fieldwork locations are the anchors that bind the data types together based on the location of the vegetation plots.

The SiDroForest data collection provides these forest inventory data such as tree height, tree crown diameters, tree species and tree stand density. The data acquired in the field span from forest inventories at the species level, tree height information and density for each vegetation plot derived from in situ assessments, to UAV camera data of the field plots that were processed to structure from motion (SfM) point clouds that give structural forest information, UAV-orthoimages of the plots, and automatically extracted tree-crowns in the form of shape file products. The individual tree-level data labelling provides already

opportunities for further machine learning applications. On top of these state-of-the art forest inventory data and UAV products that are already enriched by labelling, we prepared two data sets that can be directly used for machine learning in remote sensing applications. One data set is a synthetically generated image data set on tree crowns in the common objects in context (COCO) format (Lin et al., 2013) that we constructed from selected UAV plot data. The other data set fit for machine learning contains labelled Sentinel-2 (S-2) image patches covering the vegetation plots related to the vegetation composition. These

labelled S-2 image patches can, e.g., be used for machine learning training for a forest land cover classification using S-2 satellite images.

Photogrammetric UAV-borne products (i.e., SfM point clouds, digital elevation products, RGB orthomosaics) that extend the ground-based inventories, have already a long application history in forestry and well-defined methodological standards (e.g., Jensen et al., 2016; Panagiotidis et al., 2017). Currently, the use of UAVs in environmental applications is undergoing a

growing use in forestry and environmental science in general because of the landscape-level potential, the flexibility of the data generation and low costs (Fraser et al., 2016). Methods for the product generation of forest structure data from UAV imagery are advanced. For example, automatic tree-crown detection in UAV Red–Green–Blue (RGB) imagery has become increasingly popular in the world of computer vision and machine learning. For the United States, the National Ecological Observatory Network (NEON; Weinstein et al., 2021) dataset containing 100 million tree crowns, derived from UAV LiDAR

data that facilitated such an analysis, was created using machine learning tools such as DeepForest (Weinstein et al., 2019). Automatic tree-crown detection can be performed with UAV data using a variety of deep learning algorithms. For example, the Mask R-CNN was used by Hao et al. (2021) to detect tree crown and canopy height of Chinese fir in a plantation in China.





The UAV data was used to obtain high-resolution images of the study site. Tree crown width and tree height of Chinese fir
was manually extracted from this UAV imagery using a combination of labelled ground-truth data and canopy height model
(CHM) information and served as validation data. However, considerable effort is needed for the validation and training data
generation and there is still a lack of usable data for image classification and instance segmentation tasks as a whole. Also,
effort is needed for the extraction of tree crowns and there is a lack of usable RGB training data for automatic tree crown
detection. The central and eastern Siberian boreal zones with their forest types are especially underserved as there are no open-
source UAV forest data available. SiDroForest addresses this gap by providing on one hand extracted tree crowns of the forest
plots of shape files, on the other hand we created from the UAV RGB photos a training data set that can be used for machine
learning for this boreal forest type of Central and Eastern Siberia. For this data set, we manually extracted tree crowns to
compose 10 000 synthetic RGB images that can be used to train machine-learning algorithms to identify larch tree crowns in
UAV RGB images in the common objects in context (COCO) format (Lin et al., 2013).

In addition, as an enhanced data set ready-to use for remote sensing applications, we provide high quality labelled S-2 image
patches with vegetation categories that contains seasonal information and can be used for upscaling purposes and machine-
learning classifications.

It is increasingly common in data science and environmental science to use multiple data types within one analysis. For
example, S-2 images and metadata, topography data, CHM, as well as their combinations, were used to predict growing stock
volume using deep neural networks in four forestry districts in central Finland (Astola et al., 2021). Another example of the
use of multiple data types in non-machine learning remote sensing is the work by Wang et al. (2020) where above-ground
biomass estimation was performed using field plots, UAV-LiDAR strip data, and S-2 imagery. In this model, the partial-
coverage UAV-LiDAR data were used to link ground measurements to S-2 data. These recent studies show the need for well-
labelled publicly available data to link the data types together and for performance testing of remote sensing algorithms. In
these studies, the testing data preparation was undertaken within the project: For example, Thanh et al (2018) compared the
performance of three common machine learning algorithms; a support vector machine (SVM), a random forest (RF) and k
nearest neighbours (K-NN) on S-2 data from Vietnam. In order to validate the performance of these algorithms, the training
data (training and testing samples) were collected based on the manual interpretation of the original S-2 data and high-
resolution imagery obtained from Google Earth and 135 labelled land cover polygons were produced This study is a good
example of manually labelled data creation for a specific task and specific research area to be able to use supervised
classification tools. The work done by Abdi (2020) shows a similar study that assesses the performance of four machine
learning algorithms for land cover classification of boreal forests. Here too, the validation and training data is manually created
to assess the performance of the algorithms.

The SiDroForest labelled S-2 image patches collection provides annotated image patches for around 60 sites. As each S-2
image patch consists of nine units (pixels) of 100 m$^2$ extent each, it amounts to around 550 annotated validation and training
units. In its current stage, the SiDroForest S-2 data collection is not published with performance testing and is by us not
considered as a benchmark data set for Remote Sensing image interpretation, (e.g. Long et al., 2020). The SiDroForest labelled


S-2 image patches collection is available as a small training and validation data set providing so far underrepresented vegetation categories, that will save future users time when attempting to classify vegetation of Central Siberian and Eastern Siberian boreal forests.

By making SiDroForest public, we aim to remedy public data scarcity on UAV data of forest plots, on tree level forest data, and specifically for annotated data for the boreal forests in Central and North-Eastern Siberia and encourage the use of the data presented here for further analyses and machine-learning tasks.

## 1.1. Study Region

The data collection we provide specifically contains tree level and plot level forest-structure data from important boreal 170 transition zones located in central and eastern Siberia that are vulnerable to climate change, these are the tundra–taiga (Chukotka) and summergreen–evergreen (Yakutia) transition zones (Fig. 1).

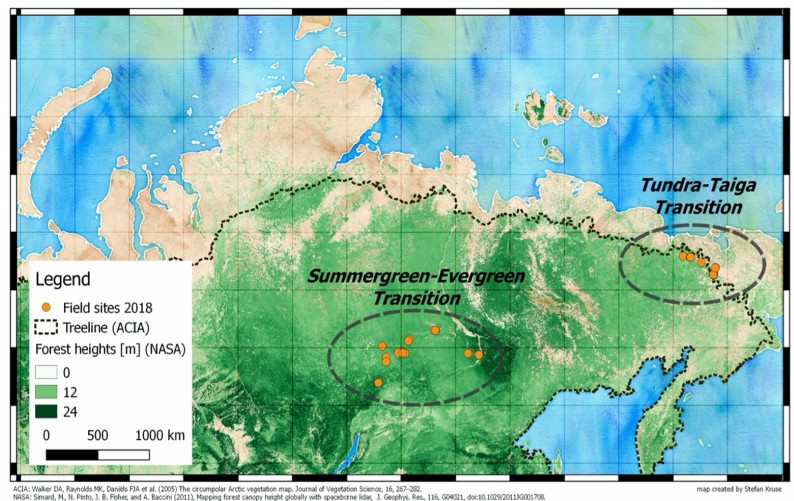

**Figure 1: Overview of the Siberian transition zones – tundra–taiga transition and summergreen–evergreen transition – covered by the 2018 Russian-German Chukotka expedition (NEFU, RU and AWI, DE). The map shows forest coverage by green colour-coded**
**forest height; the orange points are the 2018 field sites with vegetation plots.**

The tundra–taiga transition zone occurs where boreal forests reach their maximum northwards position and form a treeline ecotone (MacDonald et al., 2007). Here, the transition from open forest stands in the south and decreasing stand densities towards treeless tundra in the north takes place. A warming climate drives the transition from tundra in the tundra–taiga transition zone to open taiga forests (Rees et al., 2020). During the snow-covered season, the taiga has a lower albedo than 180 tundra due to the trees that emerge above the snow. A change from tundra to taiga albedo can result in a positive feedback loop of vegetation change which, in combination with the warming climate, may lead to dramatic environmental changes in the Arctic (Bonan, 2008). Remote-sensing data have been previously used to assess vegetation dynamics and their changes in Chukotka. Through vegetation monitoring using Landsat satellite data, Shevtsova et al. (2020) report that shrubification has expanded by 20% in area in the tundra–taiga zone and by 40% in the northern taiga as well as tree infilling occurring in the



northern taiga. Extensive satellite remote-sensing work was done by Montesano et al. to assess the vegetation dynamics in Siberia using LiDAR and synthetic aperture radar data (2014) and Landsat satellite data (2016). To be able to expand on these satellite-derived remote-sensing findings, in-depth monitoring at a vegetation plot level in this region is important. Clear overviews of species distribution over the varying types of land cover are useful to study the impacts of climate change on the eastern Siberian treeline that is not yet well enough studied, in part due to sparse data being available for the region (Shevtsova

et al., 2021). Our open-access data collection will greatly improve insights into the tundra–taiga transition zone.

Yakutia hosts the second relevant transition zone included in the SiDroForest data collection: the summergreen–evergreen transition zone. Summergreen species on the plots covered in SiDroForest consist of two species of larch trees: *Larix gmelinii* and *Larix cajanderi*. The evergreen species present are pine and spruce: *Pinus sibirica* (Only at Lake Khamra plots)/*Pinus sylvestris* and *Picea obovata*. In forests, the light-demanding summergreen *Larix* trees are outcompeted by evergreen tree taxa

(Troeva et al., 2010). Yet it is an open question as to how *Larix* forests, once established, hinder their replacement by evergreen forests and thus maintain a vegetation–climate equilibrium (Mamet et al., 2019). This self-stabilisation that takes place in the *Larix*-dominated forests in central and eastern Siberia most likely results from a combination of unique climate drivers for the region, such as vegetation, climate, fire, and permafrost (Herzschuh et al., 2020). Datasets such as the one presented here are a snapshot of the current state that can be used to monitor individual trees over time to gain a clearer idea about the vegetation

dynamics of the region.

**2 Data collection and methods**

The SiDroForest data collection contains a variety of data types that were selected to create the most comprehensive insights into the boreal forests in Siberia. Each data type has been enhanced to best use the data for vegetation-related analyses. Datasets three and four have been prepared to be used in machine-learning tasks. The combined data types aim to provide a clear picture

of the current state of the vegetation cover in central Yakutia and Chukotka.

The SiDroForest data collection is divided into four datasets (Fig. 2):

1. UAV based SfM point clouds, point-cloud products, and orthomosaics from UAV image data of expedition vegetation plots (orange hexagon symbols).

2. Individual labelled trees surveyed during the fieldwork expeditions, including information on height, tree crown, and

species (green shield symbols). These tree-individual labelled point and polygon shape files were generated and are linked to the UAV RGB orthoimages of the expedition vegetation plots.

3. Synthetically created Siberian larch tree crown dataset of 10 000 instances in Microsoft's common objects in context (COCO) format (purple triangle symbols). The images and masks contain the tree crowns of two species of larch (*Larix gmelinii* and *Larix cajanderi*) manually extracted from selected RGB UAV images.

4. Sentinel-2 Level-2 bottom of atmosphere labelled image patches with seasonal information covering the vegetation plots (red shape symbol).

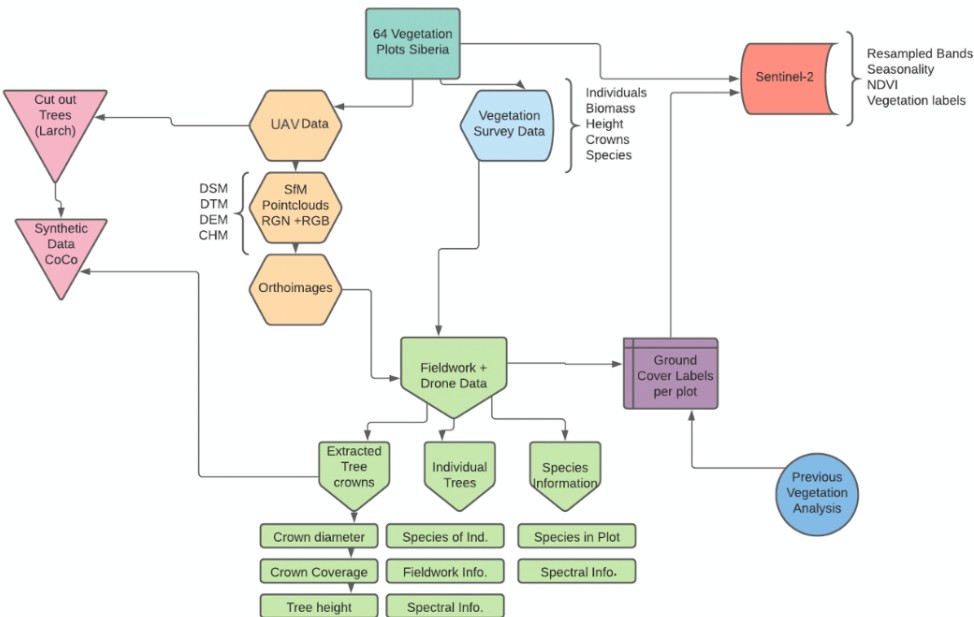

**Figure 2: Overview of the four datasets and their content and interconnections in SiDroForest.**

## 2.1 SiDroForest field data

The fieldwork data include plant taxa and forest-structure inventories from a total of 64 vegetation plots, with 39 vegetation plots in the tundra–taiga transition zone in central Chukotka and 25 vegetation plots in the summergreen–evergreen transition zone in central Yakutia. All data types included in this dataset are linked to each other using a two-letter code signifying the subregion (Table 1) and the vegetation plot numbers.

The AWI and NEFU expeditions in summer 2018 covered both the tundra–taiga transition zone in eastern Siberia and the

summergreen–evergreen transition zone in central Yakutia (Fig. 1). They covered a bioclimatic gradient ranging from treeless tundra via extremely open larch forest with mean tree heights around 5 m close to Lake Ilirney in central Chukotka (tundra–taiga ecotone) in north-eastern Siberia to dense mixed tree species stands near Lake Khamra in south-western Yakutia (boreal forest biome). The larger regions were subdivided into 12 subregions that were named based on the nearest city or lake to the plots. In Chukotka we defined three subregions encompassing 39 plots (Fig. 3a) and nine subregions encompassing 25 plots

for central Yakutia (Fig. 3b). The vegetation plots have different larch tree cover: from treeless tundra to open larch forests on slopes and in lowlands, with tree density depending on slope and slope aspect.





**Table 1: Overview of vegetation plots per transition zone, region, and subregion along with the subregion codes.**

| Transition zone | Geographical region | Subregions | Subregion codes | Plot name |
|---|---|---|---|---|
| Taiga to tundra transition zone | Central Chukotka | Bilibino<br>Lake Ilirney<br>Lake Rauchuagytgyn | BI<br>LI<br>LR | EN18000; 18028-35 (n = 9)<br>EN18001-18027 (n = 25)<br>EN18050-18055 (n = 5) |
| Summergreen to evergreen transition zone | Central Yakutia | Yakutsk<br>Magaras<br>Vilnuyi<br>Nyurba<br>Suntar West<br>Suntar<br>Mirny<br>Mirny-Lensk<br>Lake Khamra | YA<br>MA<br>VI<br>NY<br>SW<br>SU<br>MI<br>ML<br>LK | EN18061 (n = 1)<br>EN18062 (n = 1)<br>EN18063-66 (n = 4)<br>EN18067-70 (n = 4)<br>EN18071 (n = 1)<br>EN18072-74 (n = 3)<br>EN18075-76 (n = 2)<br>EN18077-78 (n = 2)<br>EN18079-83 (n = 5) |


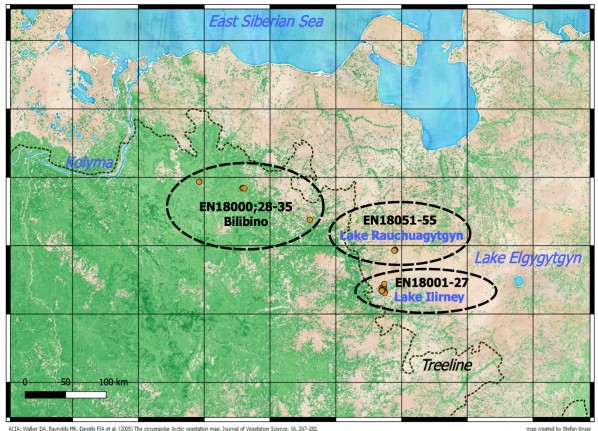

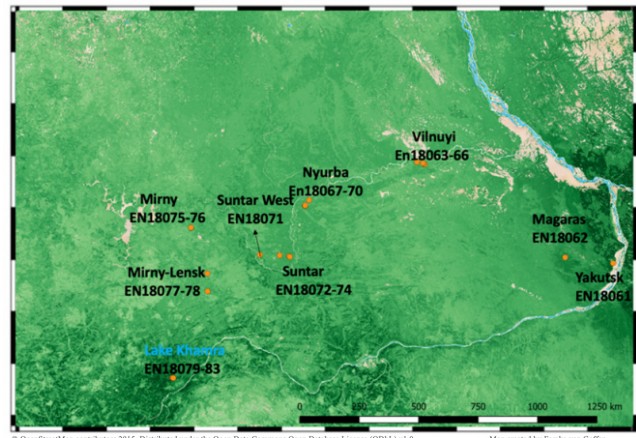

**Figure 3a. Subregions and plots for Chukotka. Bilibino (BI) (EN18000, 18028–35), Lake Ilirney (LI) (EN18001–27), and Rauchuagytgyn (RA) (EN18051–55). See also Table 1.**

**Figure 3b. Subregions and plots for Yakutia. Yakutsk (YA) (EN18061), Magaras (MA) (EN18062), Vilnuyi (VI) (EN18063-66), Nyurba (NY) (EN18067–70), Suntar West (SW) (EN18071), Suntar (SU) (EN18072–74), Mirny (MI) (EN18075–76), Mirny-Lensk (ML) (EN18077–78) and Lake Khamra (LK) (EN18079–83). See also Table 1.**

A detailed vegetation inventory was conducted for each of the plots visited during fieldwork. Fifteen-metre radius circular
plots for the projective cover of trees and tall shrubs (see Appendix Fig. A1) were set within 30 m x 30 m rectangular vegetation



plots for ground projective cover of vegetation taxa. The plots and the field data collection are described in further detail in Shevtsova et al. (2019, 2020a,b,c, 2021).

Two 30-m-long tape measures were laid out along the main cardinal directions, intersecting in the plot centre, marking the main axes of a circular area with a radius of 15 m (Appendix Fig. A4). A minimum of ten individuals of each tree and shrub
species present were selected per plot. For each individual tree we measured the stem diameter at breast height and at the base. The tree crown diameter, tree height, and vitality were estimated (Brieger et al., 2019). For Plot EN1814 and EN1865, all trees were recorded. Plot EN18070 was recorded differently to the other plots. Here, the area covered in the vegetation plot was not 15 m x 15 m but a transect with three segments: edge, transition, and centre.

We assigned a vegetation class to each plot that was determined based on the previous work by Shevtsova et al. (2020a) for
Chukotka and a principal component analysis (PCA) and tree density information from the UAV data for the Yakutia plots. In addition, the recorded species information per plot was taken into account when assigning the classes (Appendix Fig. A2 for Chukotka and Appendix Fig. A3 for Yakutia). A framework of 11 vegetation classes that can be applied to all the 64 plots and are further described in the results section was derived.

In addition to the field vegetation and the specific forest inventories that were obtained, 60 of the 64 vegetation plots were
overflown with a consumer grade DJI Phantom 4 quadcopter carrying a MAPIR Survey 3W Red–Green Near-infrared (RGNIR) camera to obtain spatially mapped detailed forest structure information in 2 and 3 dimensions (2D, 3D). The UAV imagery covered a minimum areal extent of 50 m x 50 m over the 15 m radius and 30 m x 30 m vegetation plots with a standardised flight plan following a double-grid in near-nadir position and a circular flight facing the plot centre at take-off elevation (Appendix Fig. A4). Further details are described in Brieger et al. (2019).

**2.2 SiDroForest dataset 1: Structure from motion (SfM) point clouds, point-cloud products and orthomosaics**

**2.2.1 SfM point clouds of the plots**

Due to the availability of multiple overlapping images from different camera viewpoints, point-cloud processing and the generation of 3D products and successive generation of orthoimages were possible. We manually rejected images that had been taken during take-off and landing, as well as under- or over-exposed images, from further processing (see also Brieger et
al., 2019). The remaining images were used to generate the 3D SfM point clouds and related products directly from the point-cloud data.

SfM point clouds were constructed with Agisoft PhotoScan Professional (Agisoft, 2018) according to methods described in Brieger et al. (2019). Tracked Global Positioning System (GPS) information was automatically integrated into the images during this process. The parameters were tuned with the highest resolution settings to capture as much detail of the complex
tree structures as possible. The depth filtering in the dense cloud generation was changed from the default to a mild filtering to preserve more detail especially in tree crowns (details in Brieger et al., 2019).



The dataset contains RGB and RGNIR point clouds (Fig. 4). The RGB point clouds have been segmented into two separate point clouds with a cloth simulation filter (CSF) as described in Brieger et al. (2019). The result is two RGB point clouds; one containing the ground points and very low vegetation (named here 'groundonly') and one containing the points of the higher vegetation (named here 'treesonly') (Fig. 4).

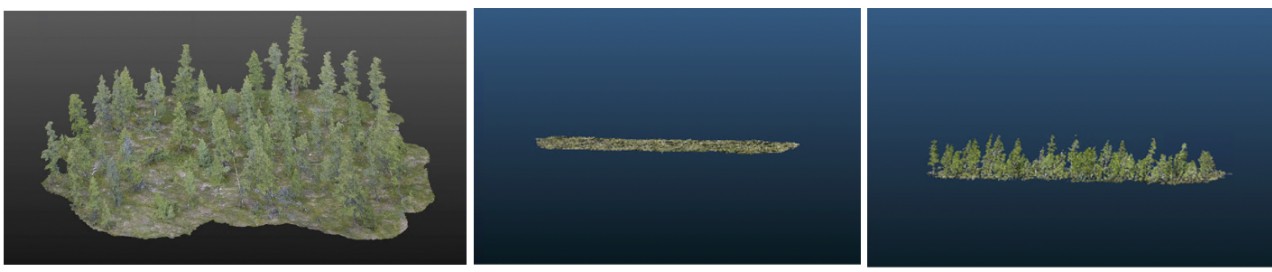

**Figure 4: Left: example of a full red–green–blue (RGB) structure from motion (SfM) point cloud for plot EN18074. Centre: segmented RGB point cloud containing only the points of the ground layer named *groundonly*. Right: segmented RGB point cloud with the above-ground vegetation named *treesonly*.**

We chose to present the point clouds segmented into these two point clouds, 'groundonly' and 'treesonly', because it reduces the size of the individual point clouds and it is easy for users to merge them together. It can also be interesting to have the two segmented when attempting to analyse the below-canopy vegetation or ground-cover classes. Plots with dense vegetation such as EN18077 and EN18063 could not be segmented into *groundonly* and *treesonly* due to the ground not being visible in the images. The point clouds are provided in the standard LASer (LAS) binary file format. The dataset includes three point-cloud types per plot: *treesonly* and *groundonly* in RGB and the full point cloud in RGNIR.

### 2.2.2 Point-cloud products of the plots

The following section describes the data products that were created from the RGB point clouds: a digital terrain model (DTM), a digital surface model (DSM), a canopy height model (CHM), and a digital elevation model (DEM). All point-cloud products were generated from the point clouds generated in Agisoft PhotoScan Professional ©. The point-cloud products were then produced in R (R Core Team, 2020) and exported as georeferenced geotiff raster files at 3 cm x 3 cm pixel resolution in the Universal Transverse Mercator (UTM) projection.

During the creation of the DEM, the choice was made to crop it to a defined area in the form of a polygon (called here the *outer polygon*) due to the better quality of the points within this region. The *outer polygon* is the area covering the camera positions plus a buffer of 5 m. In addition to the clipped product versions and the shapefiles of the *outer polygon*, the fully covered area not clipped to the *outer polygon* is also supplied for the orthomosaics and the point clouds to give the user a dynamic dataset to work with.





**DTM**

The definition of a DTM is that the surface represents the ground level with all natural and built features removed. To create the DTM for each plot, the full point clouds were separated into two subsets: the higher vegetation (*treesonly*) and the ground cover and lower vegetation (*groundonly*) using a Cloth Simulation Filter (Zhang et al., 2016) and the ground points rasterised and a minimum moving window filter applied to generate the DTM, as described in detail in Brieger et al. (2019). The SiDroForest DTM in 3-cm grid resolution represents the top of the canopy of the lowest vegetation canopy layer in case of low-structure vegetation.

**DSM**

The definition of a DSM is that the surface represents the highest-level elevation including natural and built features. The DSM is interpolated between the highest points in each grid cell representing the top of the highest vegetation canopy layer. DSMs were created in R for all available plots with a 3-cm-resolution grid used to rasterise the full point cloud, where the elevation value of the highest point in each cell is assigned as the new raster value (Brieger et al., 2019).

**CHM**

The 3-cm resolution CHM is the difference between the DSM and the DTM (CHM = DSM – DTM), and thus normalises the DSM to the ground. Each pixel represents the vegetation height above the ground (Brieger et al., 2019). The CHM represents the original point-cloud data. Because the CHM is derived from a subtraction of the DSM and the DTM, it may contain no data values where the tree crown covers a large amount of ground and the ground data are missing due to this coverage.

**DEM**

The DEM is a quantitative representation of the elevation of Earth's surface. The SiDroForest DEM provides the terrain relief referenced to the vertical datum of the World Geodetic System 1984 (WGS84) without the lowest canopy layer in contrast to the SiDroForest DTM that contains the lowest ground vegetation layer. The 3-cm resolution DEM was cropped to the *outer polygon*.

**2.2.3 Orthomosaics of the plots**

From each full RGNIR and RGB point cloud and overhead photo images orthomosaics were created. Aerial orthomosaics are geometrically corrected images that are georeferenced including the distortions by topography (the relief) and vegetation (the top-of-canopy elevation). The orthomosaics were constructed from the multiple RGB and RGNIR overhead photo images corrected for perspective and scale with Agisoft PhotoScan Professional (Agisoft, 2018) using structure from motion/multi-view stereo (SfM-MVS) techniques (Brieger et al., 2019). The RGNIR orthomosaics have been co-registered to the RGB point



clouds using the ground control points (GCPs) distributed in the field to make the RGNIR and RGB point clouds align. The orthomosaics were exported from Agisoft PhotoScan Professional (Agisoft, 2018) as georeferenced geotiff raster files at 3 cm

x 3 cm pixel resolution in the respective Universal Transverse Mercator (UTM) zone projections.

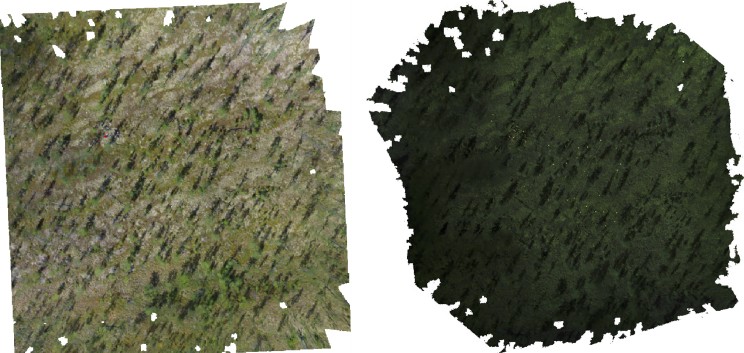

**Figure 5: Red–green–blue (RGB; left) and red–green near-infrared (RGN; right) orthomosaics for plot EN18000.**

Not all RGB orthomosaics have the same quality, as varying flight plans or weather conditions affected the construction of the final products. This resulted in 'blurry' parts in the orthomosaics. The affected plots are: EN18030, EN1878, and EN1879.

The blurry regions affect less than 20% of the image and the rest is unaffected, therefore the orthomosaics of these plots are included in the data publication.

### 2.2.4 Automated extracted tree crown polygons

The polygons of the tree crowns were created within an automated extraction workflow. The tree crowns were captured in the CHM by watershed segmentation analysis using the R package ForestTools (Plowright, 2019) and successive automatic

generation of a polygon around them following Brieger et al. (2019). This automated tree-crown detection algorithm was run for all plots and the resulting shapefiles are provided for each plot that contained trees. Quality assurance was performed for each plot by carefully examining each plot based on expert knowledge and assigning a quality score of Q1 (good quality), Q2 (medium quality), or Q3 (poor quality) to the shapefile products.

### 2.3 SiDroForest dataset 2: Individual labelled trees

In order to make assumptions and predictions about the content of the vegetation plots it is important to link the labelled individual trees from the fieldwork to the processed orthoimages. This also allows future monitoring of the individual trees and the contents of the recorded vegetation plots at large. We produced an individual labelled trees dataset containing 872 trees and shrubs that were surveyed in Siberia during the two-month fieldwork expedition in 2018 (Kruse et al., 2019). The individual species were from within the 15-m-radius vegetation survey plots and measurements such as height and crown

diameter were collected.



The individuals that could be located in the orthoimages were marked in a *point* and *polygon* shapefile that outlines the tree crown of the individual tree, containing the individual number of the tree, the species, and its form (Tree or Shrub). The form attribute was added because in the Chukotka plots there are *Pinus* species that are shrubs which can be misleading. The tree ID, exemplified in Figure 6, is the first letter of the genus of tree and the total number of individuals recorded (e.g., L259 is

the 259th *Larix* specimen). The total number of *Larix* recorded is a cumulative number over all the plots recorded. The individual number was recorded during fieldwork and corresponds to information stored in the extensive database of Kruse et al. (2020) containing measurements concerning the individual tree, which are now also accessed via the SiDroForest dataset. The point shapefiles also include the geographical x and y coordinates of the point in decimal degrees. The individual number can be used to link the tree or shrub to the rest of the information collected during the expedition such as tree height, crown

diameter, and vitality. This information is provided in a csv file in Kruse et al. (2021a).

In addition to the two shapefiles that are linked to the individually recorded trees, another shapefile is provided per plot with species-level information (Fig. 6). It contains a minimum of ten labelled polygon shapefiles that cover trees or large shrubs (>1.3 m height). These labelled polygons only cover the inside of the tree or shrub to minimize noise from the ground layer for classification purposes. For the species polygon, trees and shrubs that were seen in the rest of the orthoimages were also

included, not only the individuals included in the fieldwork records.

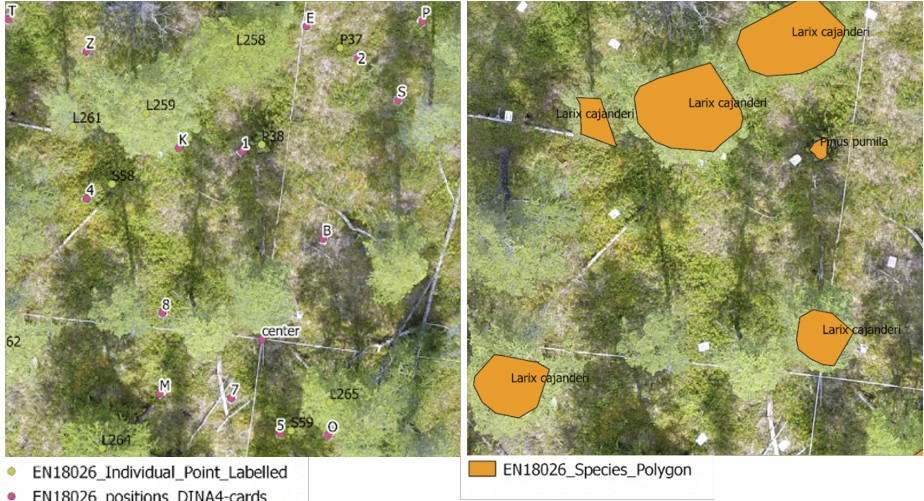

**Figure 6: Left: examples of the individual labels and right: examples of species polygons. The white bordered letters and numbers were used to mark the individuals in the field. Both shapefiles are visualised on the red–green–blue (RGB) orthoimage of plot**
**EN18026.**



### 2.4 SiDroForest dataset 3: Synthetic larch tree crowns

This synthetic Siberian larch tree crown dataset was created to enhance the data collective for upscaling and machine-learning purposes. The synthetic dataset contains larch (*Larix gmelinii* (Rupr.) Rupr. and *Larix cajanderi* Mayr.) tree crowns extracted
from the onboard camera RGB images of five selected vegetation plots from fieldwork, placed on top of fully-resized images from the same UAV flights.

To create the dataset, backgrounds and foregrounds were needed. The RGB images included for the backgrounds were from the field plots: EN18062 (62.17° N 127.81° E), EN18068 (63.07° N 117.98° E), EN18074 (62.22° N 117.02° E), EN18078 (61.57° N 114.29° E), and EN18083 (59.97° N 113° E), located in central Yakutia, Siberia (Fig. 7).


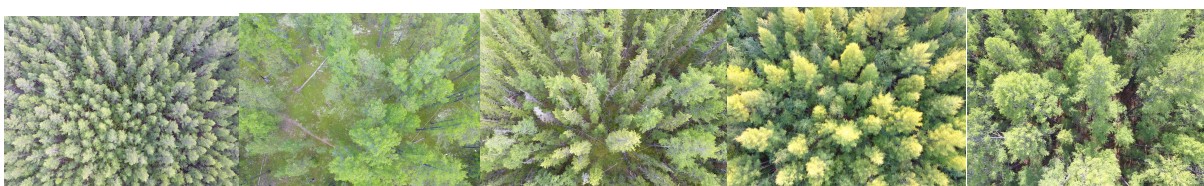

**Figure 7: Examples of red–green–blue (RGB) images of plots from the selected unmanned aerial vehicle (UAV) flights in the following order: EN18063, EN18068, EN18074, EN18078 and EN18083.**

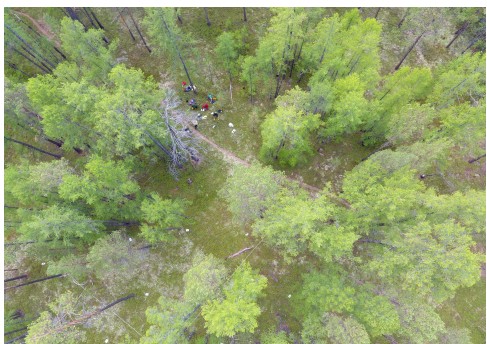

**Figure 8: Example of a red–green–blue (RGB) image that was excluded from the 35 images for plot EN18068.**

The plots were selected based on their vegetation content and their spectral differences, as well as UAV flight angles and the clarity of the UAV RGB images. For each plot, 35 images were selected in order of acquisition, starting at the fifteenth image in the flight to establish the backgrounds for the dataset. The first fifteen images were excluded because they often contain a visual representation of the research team (Fig. 8).

The raw UAV RGB images were cropped to 640 by 480 pixels at a resolution of 72 dots per inch (dpi). These are later rescaled to 448 by 448 pixels in the process of the dataset creation. In total there are 175 cropped backgrounds.

The foregrounds used in the dataset consist of 117 tree crowns and were manually cut out using Gimp V2.10 software (Gimp, 2019) to ensure that they were all *Larix* trees (see Fig. 9). Of the tree crowns, 15% from the margins of the image were included to make sure that the algorithm does not rely on a full tree crown in order to detect a tree.

The COCO format for the SiDroForest synthetic dataset is stored in a JavaScript Object Notation (JSON) file that contains the mask and image name, the colour category that was used to create the mask (Appendix Fig. A5), the category the image falls under, which in this case is 'larch' and the super category which is 'tree'. This way the created masks are connected to the created images.

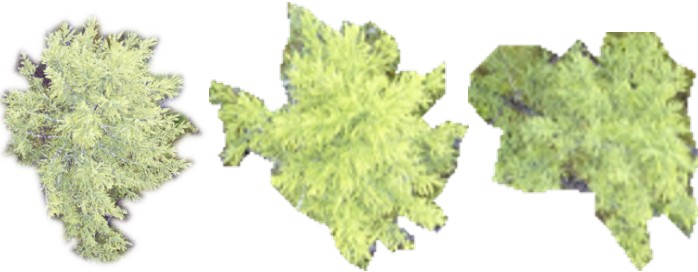


**Figure 9: Examples of cut out tree crowns.**

The extracted tree crowns were rotated, rescaled, and repositioned across the images using the cocosynth algorithm developed by Kelley (2009) resulting in a diverse synthetic dataset that contains 10 000 images for training purposes and 2000 images for validation purposes for complex machine-learning neural networks. In addition, the data are saved in the Microsoft COCO

format (Lin et al., 2013) and can be easily loaded as a dataset for networks such as the Mask R-CNN, U-Nets, or the Faster R-NN. These are neural networks for instance-segmentation tasks that have become more frequently used over the years for forest monitoring purposes.

**2.5 SiDroForest dataset 4: Sentinel-2 satellite image patches**

Sentinel-2 (S-2) is an ESA optical satellite mission providing satellite imagery globally and is freely available, which facilitates

low-cost broad-scale analyses of circumpolar boreal forests. The S-2 mission is composed of two identical satellites that were launched in 2015 and 2017 (ESA, 2015). The S-2 imagery has 13 multispectral bands, where in the native spatial resolution four bands have the highest (i.e., 10 m) spatial pixel resolution covering the visible wavelength region with three spectral bands (blue, green, red), and one spectral band in the near infrared (NIR). An overview of the S-2 spectral bands can be seen in Appendix Table A1.

The best possible acquisitions of S-2 data, that is, cloudless and without smoke from forest fires, were retrieved from the ESA archive from the years 2016 to 2020 for three distinct time stamps: early summer (May to June, depending on latitude), peak summer (mid-July to early August), and late summer (late August to September). The ESA S-2 L1C (top of atmosphere) image data were processed to Level-2A (bottom of atmosphere) surface reflectance using the newest version of the atmospheric correction processor Sen2Cor (ESA Sen2Cor, 2020). Atmospheric correction processing was performed mainly with the

default configuration which uses a rural aerosol model with a start visibility parameter of 40 km corresponding to aerosol optical thickness of 0.20 at 550 nm. Actual aerosol optical thickness is determined during the atmospheric correction



processing. The only two non-default settings were the use of the Copernicus DEM for terrain correction (Copernicus, 2021) and the use of vertical column ozone content from L1C-metadata instead of a fixed value of 331 Dobson units.

The data provided in SiDroForest are optimised for vegetation-related analyses, such as resampling the bands to 10 m spatial
resolution to make them comparable at the same resolution and removing the 60 m bands that support atmospheric correction but are not optimal for land surface classification. The dataset presented here contains subsets of S-2 acquisitions that cover all the 64 locations where fieldwork was performed in Siberia in 2018. The spectral Bands 5/6/7/8A/11/12 were resampled from 20 m to 10 m.

The NDVI was calculated using (B8 – B4) / (B8 + B4) and masked for surface waters using the water-mask provided with the
L2A-product. Areas of snow and lake and river ice in early season acquisition NIR bands were masked using an adaptable optimised threshold. For each site, three time-stamps were included and an NDVI band was added (Fig. 10).

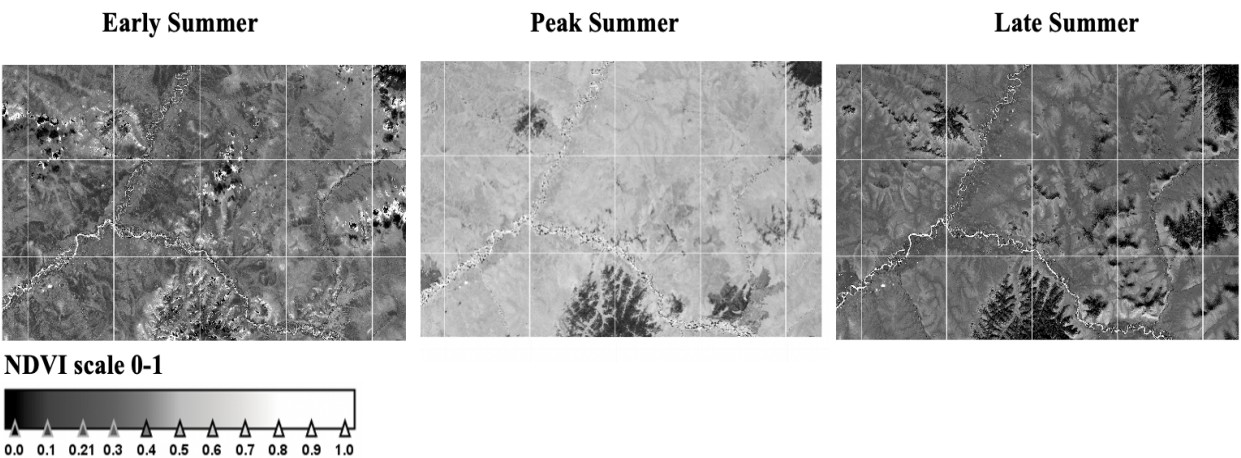

**Figure
10: Sentinel-2 NDVI in greyscale for the three-time stamps for Bilibino, Chukotka. Left: early summer, centre: peak summer, and right: late summer.**

The pre-processed S-2 imagery with the spectral bands 2,3,4 (visible), 5,6,7,8A (NIR), and 11,12 (SWIR; short-wave infrared) at 10 m resampled spatial resolution and the additional water-masked NDVI band are cropped to 30 m x 30 m image patches around the centre coordinate of the vegetation plot using UTM oriented shapefiles. These shapefiles and the JSON-annotated image patches receive one of the 11 vegetation classes derived from fieldwork and analysis of the UAV data, described in section 2. 2. 1, as attributions. The labels are also stored in a JSON file for each plot in accordance with the patch labelling in
BigEarthNet-S2 (Sumbul, 2019). JSON is an open standard file format and data interchange format that uses human-readable text to store and transmit data objects consisting of attribute–value pairs and arrays. It is often used in machine learning as the standard for stored labels.





## 3. Results

### 3.1 Dataset 1: SfM point clouds and point-cloud products


The total number of RGB and RGNIR point-cloud points were compared per plot. In order to get the full number of points for the RGB point clouds, the *treesonly* and *groundonly* segmented points were added together (Fig. 11). For most of the plots, the total number of point-cloud points were of a similar magnitude, especially for the Chukotka plots. With higher vegetation structure, the NIR reflectance enables more data points in RGNIR point clouds over the high and dense Yakutian forest plots. Not all plots have all data available due to technical problems during the data collection (plots EN18035–EN18055, EN18069, EN18071, and EN18076 have no UAV data).


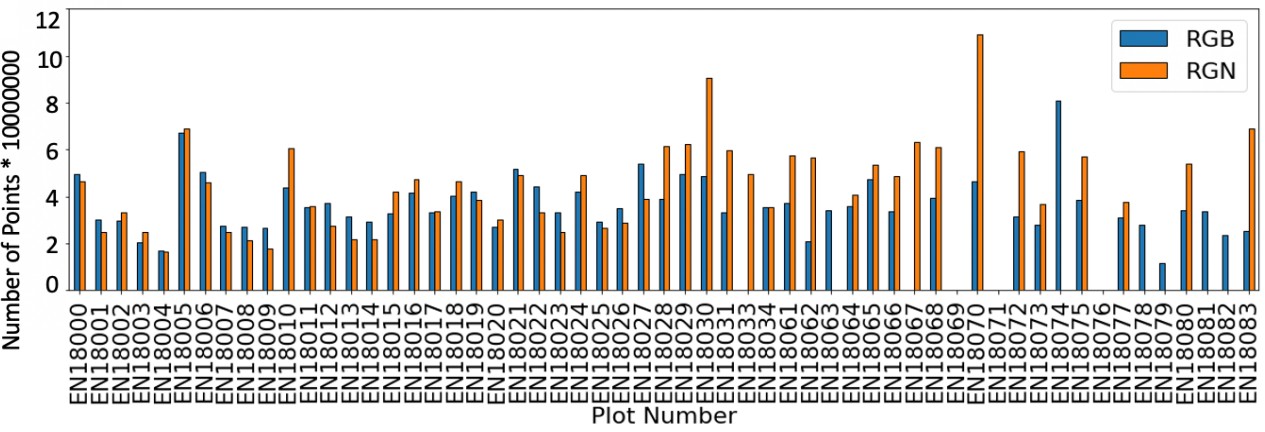

**Figure 11: Comparison of the number of points in red–green near infrared (RGN; orange bars) and the combination of the two red–green–blue (RGB) *groundonly* and *treesonly* point clouds (RGB; blue bars).**

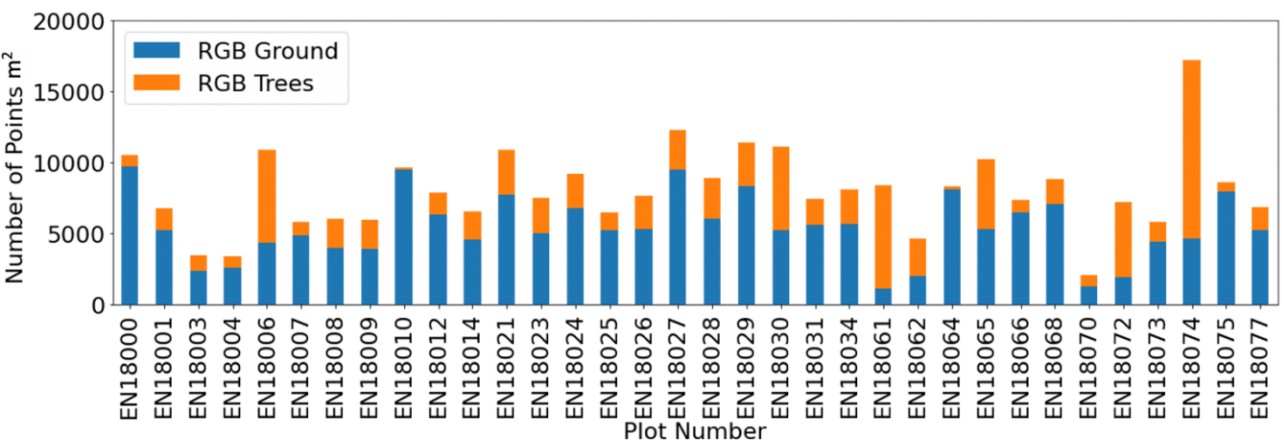


**Figure 12: Ground and above-ground points per segmented point cloud per m².**

For the segmented RGB point clouds, the ground to above-ground ratio was inspected (Fig. 12). The plots that have substantially more points in the above ground (*treesonly*) part also have more vegetation cover.



**Point-cloud products**

For each plot, the generated point-cloud products provide insights into the forest stand structure and the elevation and density
       of the plots (Fig. 13).

**Figure 13: Digital terrain model (DTM), canopy height model (CHM), digital surface model (DSM), and digital elevation model**
**(DEM) for plot EN18077.**

**Automated tree-crown polygons per plot**





The automatic tree-crown detection algorithm detected tree crowns for each plot. The tree-crown detection was not equally successful for each plot. For this reason, a quality control label (Q1, Q2, Q3) was included with every shapefile in the name. Q1 represents the best quality and with the increasing number the quality decreases (Fig. 14). Each generated tree crown also has an attribute table assigned that contains information on tree height etc. (Table A2, Appendix).

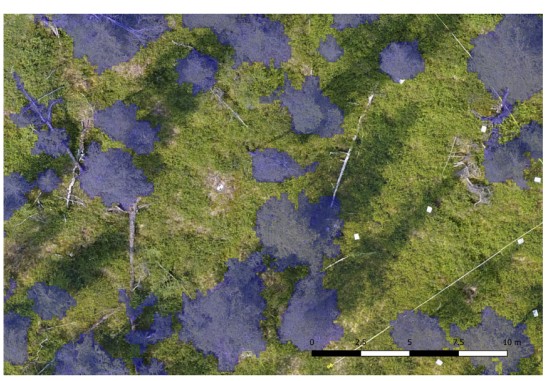
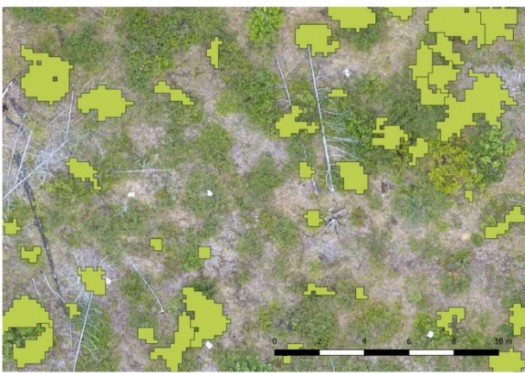

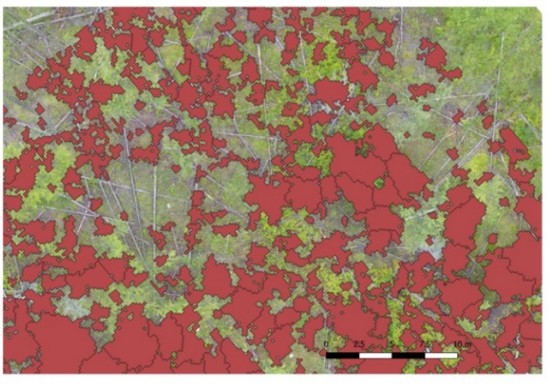

**Figure 14: Top left: Crown polygons for plot EN18014 with Q1 quality score. Top right: Crown polygons for plot EN18014 with Q2**
**quality score. Bottom: Crown polygons for plot EN18014 with Q3 quality score.**

Each plot has a different number of automatic tree crowns detected, this number depends on the density and the quality of the detected crowns in the plot. The percentage of crowns covering each plot was calculated to show the coverage of trees per plot (Fig. 15).

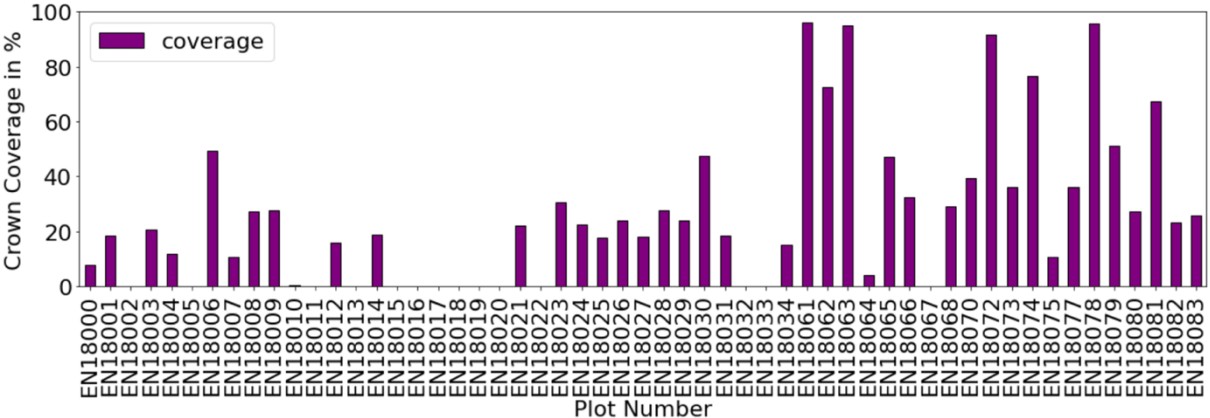

**480**    **Figure 15: The percentage of the crown coverage in the orthomosaics. A high percentage reflects a denser forest.**

**3.2 Dataset 2: Individual labelled trees**

We located 872 trees and large shrubs that were recorded during fieldwork in the orthoimages. A minimum of ten individual trees were recorded per plot with the exception of plots EN18014 and EN18028, where all trees present were recorded. The 872 trees and large shrubs recorded in the field were selected as representative individuals for their respective plots. Not all

**485**    individual trees that were recorded in the field were also located in the orthomosaics (Fig. 16). The mean tree height and mean crown diameter per plot can be seen in the Appendix (Fig. A6 and Fig. A7).

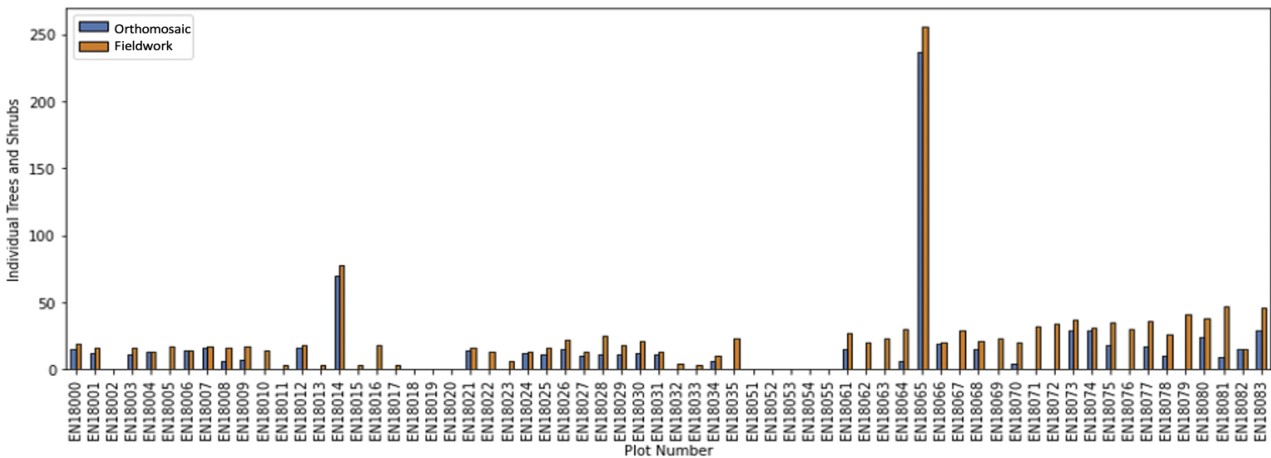

**Figure 16: Number of individual trees recorded in the field and visually identified and relocated in the red–green–blue (RGB) orthomosaics per plot. For plots EN18014 and EN18065 all trees were recorded that were present on the plot.**

**490**    For each tree or shrub from fieldwork visually identified in the orthoimages, the created point and polygon shapefiles contain information about the tree or shrub species visible in the orthoimages. For each located individual, the three shapefiles pinpoint



the location, add a unique identifier, and record the species information and are overlain on the RGB or RGNIR orthoimages of the plots as a useful visualisation (Fig. 17).

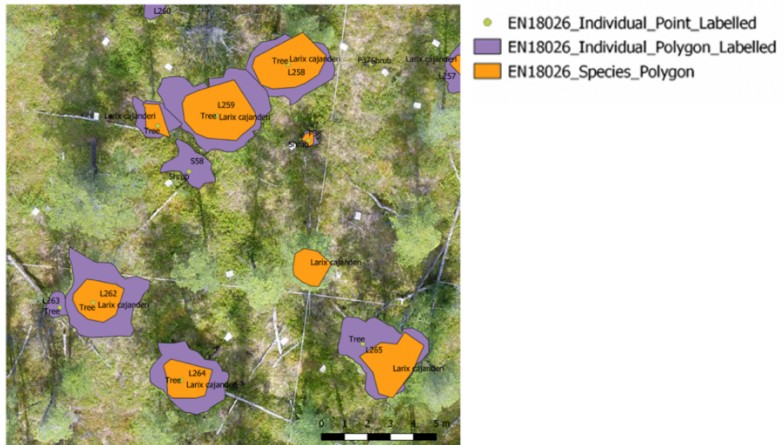

**Figure 17: Overview of the three types of shapefiles included in the individual labelled trees dataset visualised on top of a red–green–blue (RGB) orthoimage.**

### 3.3 Dataset 3: Synthetic dataset results

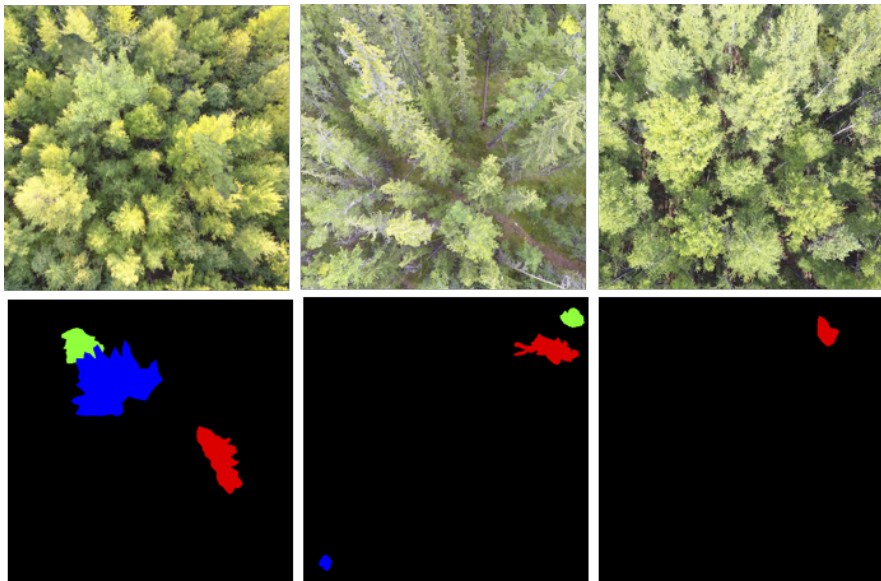

**Figure 18: Examples of synthetic images and corresponding masks generated. The images show three drone flight images with a cut out larch tree overlay. The masks below show the location of the placed trees in the form of masks. Each mask is assigned a different colour to distinguish the masks.**

Examples of the results for the synthetic larch tree crowns include the RGB images that were generated and the accompanying masks that are used for the instance segmentation and object detection tasks as shown in Figure 18. The synthetic larch tree



crown RGB image database has many different larch-dominated forest structures. This creates a large diversity of spatial and
spectral features for machine-learning tasks.

## 3.2 Dataset 4: Sentinel-2 labelled image patches

The S-2 dataset comprises 30 × 30 m labelled image patches per plot, with 11 bands and vegetation labels assigned to each
plot. The information per patch is stored in a JSON file. Figure 19 provides a schematic overview. It also includes three
seasonal time stamps: early summer, peak summer, and late summer. A visual representation of the classes can be seen in
Figure 20. In total there are 11 vegetation classes for Chukotka and Yakutia. The vegetation classes were added in the name
for each patch as well as in the JSON file. The classes and their distribution are shown in Figure 21. A table with all the class
labels assigned per plot can be seen in Appendix Table A3.

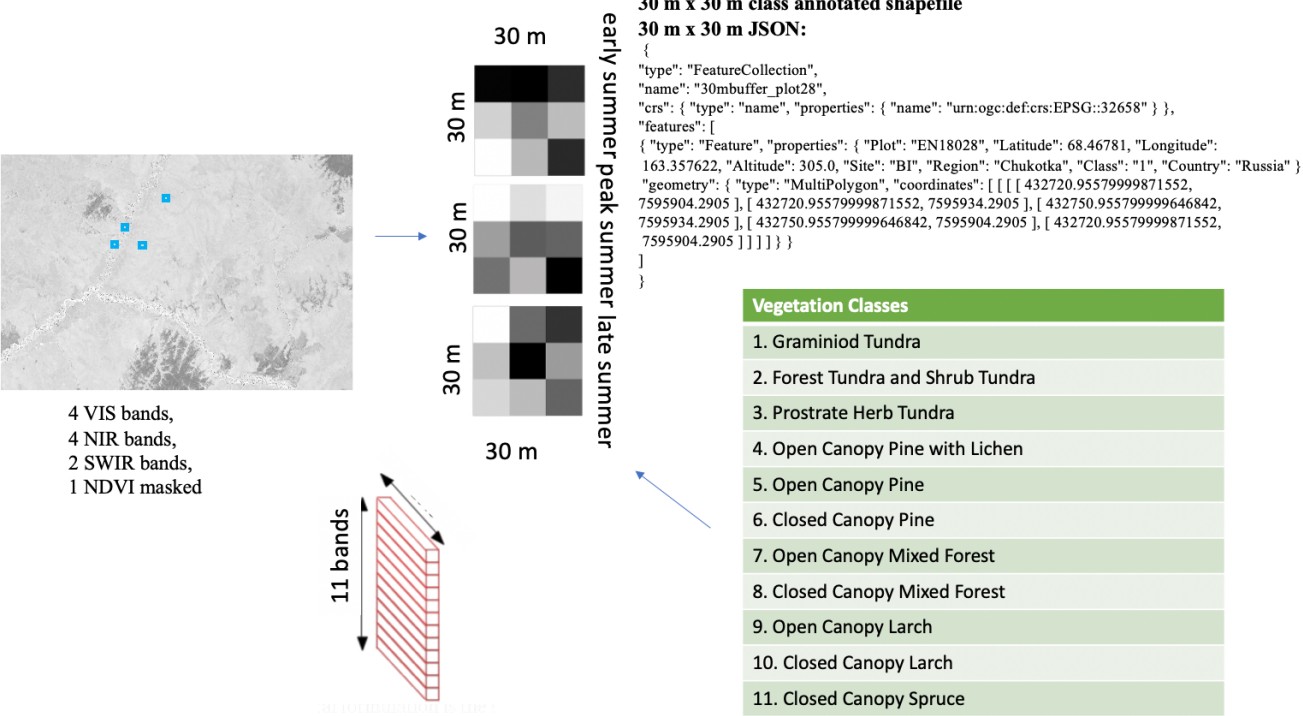

**Figure 19: Overview of the products in the Sentinel-2 data set, exemplified for plot EN18028.**



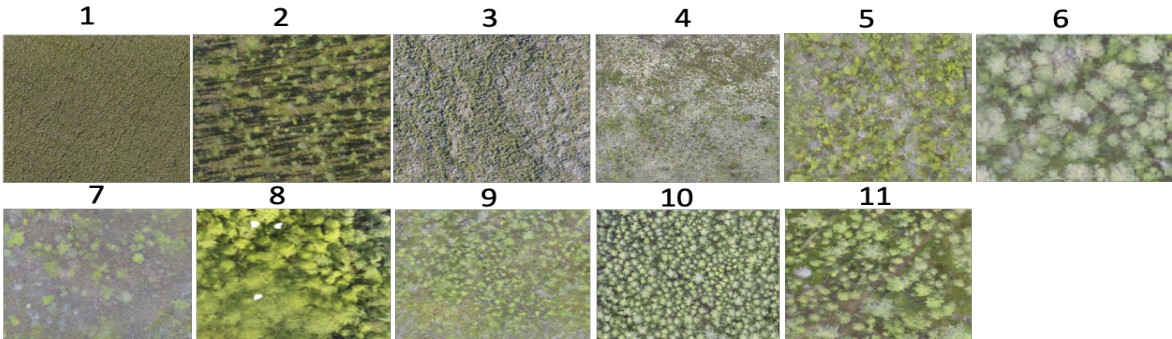

**Figure 20: Visual representation of the 11 vegetation classes from the orthomosaics (see also Appendix Table A3). The varying quality of the orthomosaics is described in 2.2.3.**

**Table 2: Vegetation class labels per plot and percentage of plots for each classification.**

| Vegetation label | Percentage of plots with this label |
| --- | --- |
| Graminoid tundra | 39% |
| Forest tundra and shrub tundra | 4% |
| Prostrate herb tundra | 21% |
| Open canopy pine with lichen | 2% |
| Open canopy pine | 2% |
| Closed canopy pine | 2% |
| Open canopy mixed forest | 10% |
| Closed canopy mixed forest | 4% |
| Open canopy larch | 4% |
| Closed canopy larch | 10% |
| Closed canopy spruce | 2% |



## 4. Discussion

### 4.1 A comprehensive data collection on Siberian boreal forests

To date, the most relevant open-source datasets available on boreal and arctic vegetation data are from the long-term ABoVE NASA Terrestrial Ecology Program, focusing on boreal and arctic regions in Alaska and Canada. The ABoVE data collections contain field-based, airborne, and satellite sensor-derived data, providing a foundation for improving the analysis and modelling capabilities needed to understand and predict climate change in the arctic and boreal regions. There are fifty vegetation related datasets published so far in the ABoVE Science Cloud (ASC): eleven thematic maps, mostly derived from

remote sensing and focused on Alaska, nine vegetation-variable related mapped remote-sensing products, mostly covering large regions, one time series product extracted for the footprint of a flux tower, and six ground-based vegetation related data collections, including data from ten terrestrial LiDAR vegetation plots (Maguire et al. 2016) and 24 vegetation plot surveys . The circum-arctic vegetation map north of the treeline (CAVM Team, 2003) is one circumarctic product, the other forty-nine datasets are all located in Alaska.

SiDroForest provides a new comprehensive data collection of a variety of data types that were selected to create the most useful insights into the boreal forests, specifically the larch-dominated forests representative of central and eastern Siberia. The focus of the SiDroForest data collection is, at this stage, not to provide thematic maps or upscaled remote-sensing products but to provide a rich, open data source on ground-based and UAV-derived information and labelled data types enhanced to best use the data for vegetation-related analyses and machine-learning tasks. The data types combined aim to provide a clear

picture of the current state of the vegetation in the tundra–taiga and summergreen–evergreen transition zones.

In this newly assembled SiDroForest data collection we provide i) labelled field data that are new data from the field studies in 2018 in Siberia for the Chukotka and Yakutia regions and ii) link SiDroForest data in the PANGAEA data repository to the already published datasets via their metadata. Publications by Shevtsova et al. (2019, 2020b) include the 2016 and 2018 vegetation inventories on the projective vegetation cover, while the total and partitioned biomass data of the 2018 vegetation

plots for the tundra–taiga transition zone in Chukotka are found in Shevtsova et al. (2020c). Kruse et al. (2020, 2021) published two forest inventories for the 2018 vegetation plots, one for the tundra–taiga in Chukotka and one for the summergreen–evergreen transition zone in central Yakutia. Brieger et al. (2019a,b) supply ten ultra-high resolution photogrammetric point clouds from the UAV overflights in 2018 over forest vegetation plots. For these ten plots, the construction of RGB SfM point clouds was evaluated and optimised and was then used to process all RGB and RGNIR SiDroForest point clouds. We provide

here the complete and comprehensive dataset of the full range of SfM-derived products (Kruse et al. 2021b) including all the RGNIR and RGB point clouds from all 60 overflown vegetation plots with enhanced field data information such as the individually labelled tree dataset (van Geffen et al., 2021b).





The SiDroForest data collection also contains 19 342 automatically detected tree-crown polygons (Kruse et al. 2021b). Bieger et al. (2019) report the comparison of field measured and detected crown diameters (mean $R^2 = 0.46$, mean RMSE = 0.673 m,

mean RMSE% = 24.9%). The weak correlation between observed and detected crown diameters is explained by Brieger et al. as being due to the quality of the available field data, which are estimations instead of direct measurements and therefore could have decreasing precision with increasing heights.

Automatic tree-crown detection has become more frequent due to the availability of state-of-the-art instance segmentation algorithms from the world of computer vision (Neuville, 2021). An example of previous work using a neural network tree-

crown detection is Braga et al. (2020), where the Mask R-CNN (He, 2017) was used to perform the tree-crown detection and delineation. This exemplifies how the synthetic dataset in SiDroForest (van Geffen et al. 2021c) could be used for analysis as the Mask R-CNN is trained with a COCO format dataset.

The format of how to provide extracted tree crowns can differ: Weinstein et al. (2021) provide a 100-million individual tree crowns dataset covering a large area and standardised LiDAR remote-sensing data from the National Ecological Observatory

Network (NEON) which is very impressive. Here a CHM was used to filter out all canopy tops over 3 metres in height from 37 different NEON sites. The individual tree crowns in Weinstein et al. (2021) are represented by a bounding box shapefile that approximates the crown area and links to the tree attributes. The SiDroForest tree-crown dataset (van Geffen et al., 2021c) provides the tree crowns in the form of a crown-delineating polygonal shape enriched with attributes. The SiDroForest tree-crown data cannot cover a comparably large area as the NEON airborne LiDAR data collection extending over 1 km x 1 km

tiles, but it does offer plot-size coverage of tree crowns with useful data for machine learning and computer vision applications. The SiDroForest tree-crown data (van Geffen et al., 2021c) are specifically made to detect Siberian larches in different mixtures of mixed summergreen needle-leaf and evergreen needle-leaf forest. Additionally, the synthetic SiDroForest dataset (van Geffen et al. 2021a) provides 10 000 images for training and 2000 validation images to train neural networks.

The SiDroForest data collection also provides labelled S-2 satellite image patches per vegetation plot (van Geffen et al. 2021b)

that can be used as ground-truth data for machine-learning classifications. Though freely available and operationally downloadable, S-2 data are not ready to use. Despite a frequent acquisition rate at higher latitudes, S-2 data often contain clouds and finding a cloud- and haze-free acquisition can take time, even with cloud filtering. It is common practice that users produce labelled patches of satellite data that function as parameterisation for classification and upscaling purposes. For example, BigEarthNet (Sumbul, 2019) is a large-scale open-source dataset that provides labelled S-2 image patches (now

called BigEarthNet-S2, previously *BigEarthNet*) acquired between June 2017 and May 2018 over ten countries. Each patch includes a JSON file with the ground cover labels for the patch. In accordance with the structure of BigEarthNet-S2, the SiDroForest image patches are also accompanied by a JSON file that contains the class labels per image patch. BigEarthNet-S2 provides patches of larger area coverage to represent 'landscapes' such as estuaries. The purpose of the SiDroForest S-2 image patches and labels lies in the true representation of vegetation classes and evergreen needle-leaf mixed forest and the

seasonal time stamps of early summer, peak summer, and late summer.





## 4.2 Importance of labelling and data quality

Labelling accurately is one of the most important aspects for a usable dataset for machine-learning purposes. If the labels are inconsistent or very uneven the classification tools will have trouble correctly identifying the classes. The SiDroForest data
collection contains a variety of labels per dataset.

The labels for the *Individual Labelled Tree* dataset (van Geffen et al. 2021c) contain information on species and location of the individual tree or shrub. These data have been verified and checked, yet in some instances two trees are located very close to each other or the location was not recorded correctly in the field and an individual tree or shrub could not be found in that case. The difference between the number of trees recorded in the field and located in the orthomosaics can be seen in Figure
15. The UAV images were inspected based on expert knowledge to locate the trees as accurately as possible. However, dense forest plots in Yakutia posed a problem for locating all the individuals correctly and not all individuals recorded in the field could be located in the orthoimages for those plots. Figure 22 shows an example of dense forest plots.

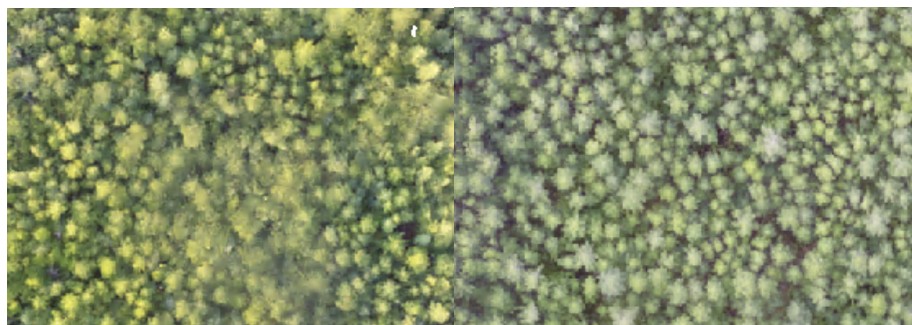

**Figure 21: Dense forest red–green–blue (RGB) orthomosaics for plots EN18077 and EN19063.**

The SiDroForest synthetic dataset (van Geffen et al., 2021a) has written labels in the JSON format (Appendix Fig. A5) that contain the higher category, or 'super category', 'Tree' and subcategory 'Larch'. The two categories exist in case there are more species added under the higher-level label 'Tree' (example in Appendix Fig. A5). The current set identifies all larch trees, regardless of which species, since the sites covered contain two larch species: *Larix cajanderi* and *Larix gmelinii*. The
two species of larch here have only the one label larch because the aim was to identify all larch trees in both Chukotka (solely *Larix cajanderi*) and Yakutia (predominantly *Larix gemelinii*). It would be an enhancement of the dataset in the future to distinguish between the two species of *Larix* in the labels as well. The dataset can be further enhanced by adding the other dominant tree species for the region: spruce and pine.

The backgrounds were carefully selected for the synthetic dataset to create diverse scenes and forest information for the
algorithm to learn from. This can help the algorithm detect larch trees on multiple backgrounds. However, it may also introduce noise into the dataset. As investigated by Xiao (2020), on one hand, there is evidence that models succeed by using background correlations but on the other hand, advances in classifiers have given rise to models that use foregrounds more effectively and are more robust to changes in the background. These findings suggest that the performance of the algorithm is more important



than the consistency of the backgrounds in a dataset. However, it is still important to be aware of such interference, and

extensive benchmarking is needed to evaluate the performance of an instance segmentation or object detection algorithm for

the dataset, which we are planning to undertake.

The dataset also contains generated RGB images that should contain natural looking scenes. In practice, not all the RGB images

look as natural as others (for example, parts of images in Fig. 23). The unnatural image construction is mostly due to variation

in size compared to the images placed on them. Since there are 10 000 images in the dataset these unnatural images do not

strongly undermine the natural ones and make up less than 10% of the total images.

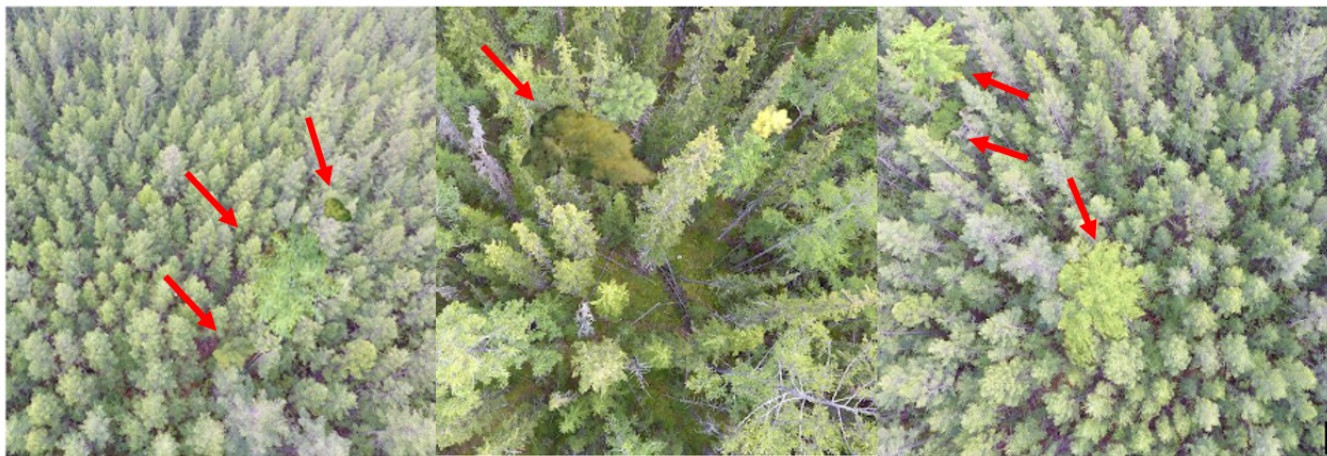

**Figure 22. Examples of unnatural looking generated images in the synthetic image dataset, the red arrows show the cut-out larch trees that were placed over the UAV images.**

The synthetic dataset could be further extended by using the labelled crowns_polygon data that are included in the individual

labelled trees dataset (van Geffen et al., 2021b) to automatically extract the tree crowns. This would generate another synthetic

tree-crown set that represents all the fieldwork individuals that were collected. This dataset would, in addition, include more

of the species population per region.

**4.3 Classification potential with Sentinel-2 and labelled image patches**

Despite the increased availability of satellite missions and open-source remote-sensing data and products, challenges remain

that are particular to terrestrial high-latitude ecosystems. Seasonal challenges such as the combination of snow cover over a

long time of the year, a short and rapidly progressing growing season, high cloud frequency, and low sun angles pose a problem

for comprehensive remote-sensing collections in the sub-Arcic region (Beamish et al., 2020). SiDroForest aims to remedy this

scarcity by providing this high-quality dataset of S-2 data linked to published field inventories (van Geffen et al., 2021b). The

final labels for the S-2 labelled patches provide ground-truth information needed to classify larger areas. The Yakutia field

data collection covered diverse plots as seen in the vegetation classes assigned (Table 2) which may pose a problem for

classification as the classes are unevenly distributed. When the fieldwork was undertaken, multiple plot sites covering different




classes were preferentially recorded in close proximity to each other for time-related reasons. The time spent in fieldwork is limited and expensive and a variety of different data can be collected close to each other. The diversity of the collected fieldwork data has advantages and disadvantages for machine learning. On the one hand it is good to have many different

vegetation types covered in the field plots to log the diversity of the vegetation cover for the region. On the other hand, more ground-truth data plots in the same category will greatly improve classification of satellite data and too much diversity in the classes hinders a balanced classification. For example, label 4: Open Canopy Pine with Lichen, only occurs in one plot. Spectrally, this plot is different from the others due to the presence of the almost white coloured lichen. It was therefore important to label this plot differently from the others, even if this creates uneven and unbalanced labels.

An idea to remedy this in the future is to add in more fieldwork plot locations from upcoming expeditions and to supplement the plots with S-2 data that are similar in their spectral signal from the same region. If the user wants to classify S-2 based on the labelled patches, it is important to keep the unbalanced classes in mind.

The classes assigned to the S-2 image patches were tested with simple machine-learning algorithms. The patches were extracted for both Yakutia and Chukotka and used together to classify all sites. A Gaussian Naive Bayes performed best with

82% overall average accuracy per class for the Yakutia sites. The preliminary results for one of the Yakutia sites are shown in Figure 23, chosen due to the diverse vegetation at the site, to show the classification potential.

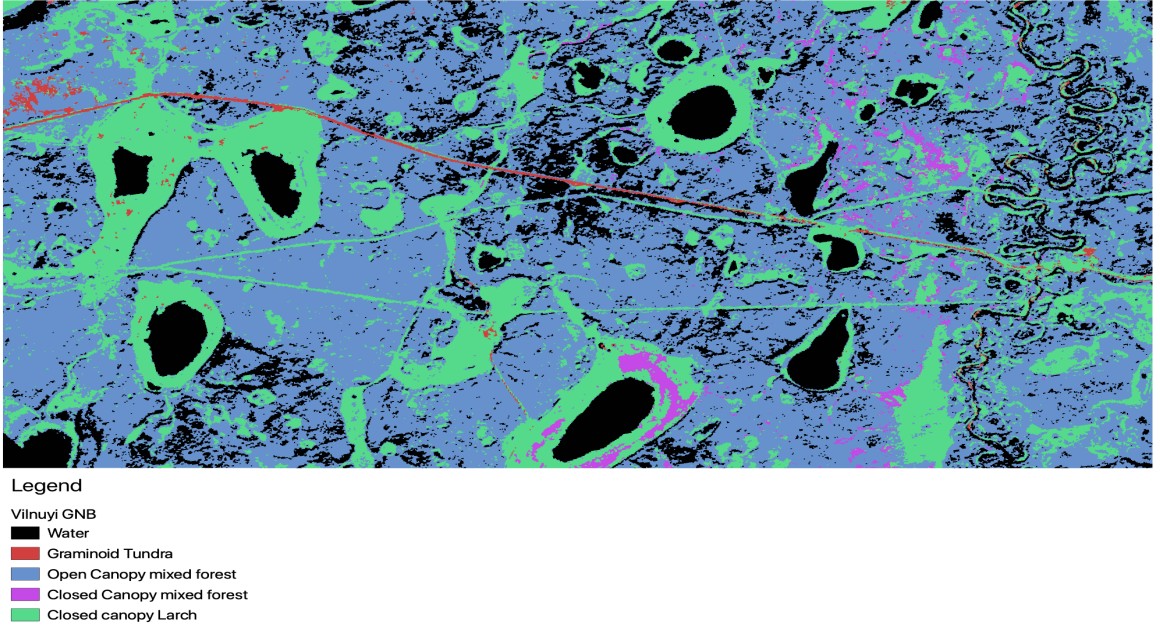

Legend

Vilnuyi GNB
- ⬛ Water
- 🟥 Graminoid Tundra
- 🟦 Open Canopy mixed forest
- 🟪 Closed Canopy mixed forest
- 🟩 Closed canopy Larch

**Figure 23: Classification of the Vilnuyi Sentinel-2 image patch based on the vegetation labels in SiDroForest.**



## 5. Conclusions

The circum-boreal forests are covering large areas on the globe. Every new forest data set collected, processed further and published in a ready-to be used format for a wide range of biological and ecological applications is therefore quite rare and an important addition for scientific studies that aim to better understand global forest dynamics.

The datasets presented here provide a comprehensive overview of the vegetation structure using a variety of data types. The fieldwork locations are the anchors that bind all the data types in this data collection together. The datasets include fieldwork information from vegetation plots and UAV acquisitions from field expeditions. The data collection spans from forest inventories at the species level, tree height information and density for each vegetation plot, UAV-derived SfM point clouds that provide structural forest information, RGB and RGNIR orthoimages from the plots, to S-2 image patches of seasonal information annotated with vegetation categories that can be used for upscaling purposes to a larger region.

Combining the data types within SiDroForest can lead to a better understanding of forest structures and vegetation dynamics. The future states of boreal forest are still largely unpredictable: labelled field data and remote-sensing data provide the tools for machine-learning based applications to help forecast likely scenarios.

The increased use of machine-learning techniques in the field of remote sensing and forest analyses call for more and better labelled data. If forest structure data are rarely available for the tundra–taiga and summergreen–evergreen transition zones, even less is available that can be used for machine learning, such as optimised data containing labelled vegetation. In addition, due to the remote nature of the dataset locations, obtaining ground-truth data is difficult and expensive. The current data collective provides as base for the data collected by AWI and NEFU during expeditions in the future. Adding consistently to the SiDroForest dataset the upcoming years, the usability and relevance of this data collective will grow with time. By making this data collection open source, we aim to remedy data scarcity on tree level forest data for the region and we encourage the use of the labelled tree level and plot level forest data sets presented here for further analyses and machine-learning tasks.

## 6. Data availability

All four data sets of the SiDroForest Data collection are published in the PANGAEA data repository and are available for download:

i) UAV-SfM point clouds, point-cloud products, and orthoimages: https://doi.pangaea.de/10.1594/PANGAEA.933263,

ii) Individual labelled trees: https://doi.pangaea.de/10.1594/PANGAEA.932821,

iii) Synthetically created tree crowns dataset: https://doi.pangaea.de/10.1594/PANGAEA.932795

iv) Sentinel-2 labelled image patches: https://doi.pangaea.de/10.1594/PANGAEA.933268





# 7. Appendix

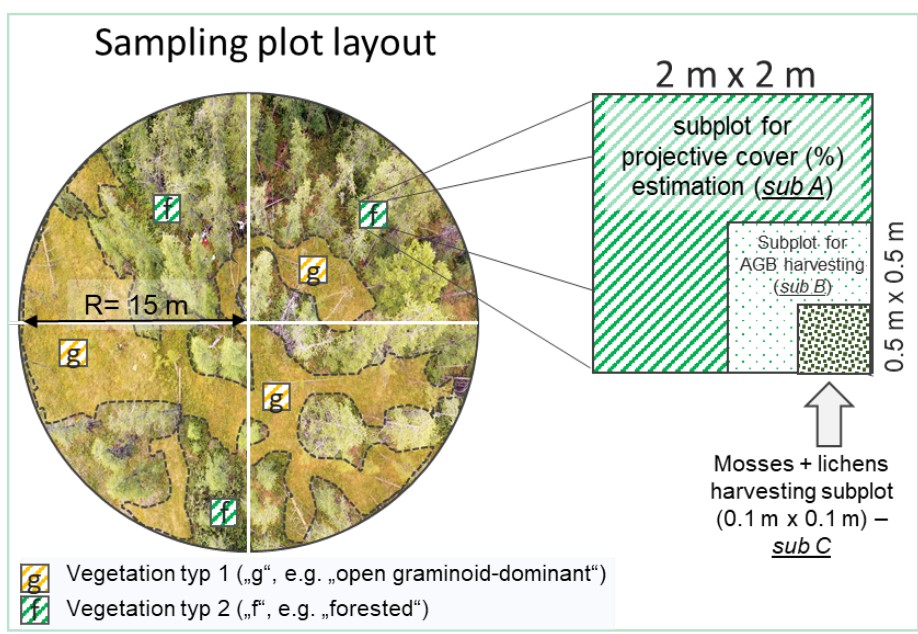

**Figure A1: Sampling scheme of the 2018 expedition vegetation survey. Projective cover of tall shrubs and trees was estimated on a circular sample plot with a radius of 15 m (after Shevtsova et al. 2020).**

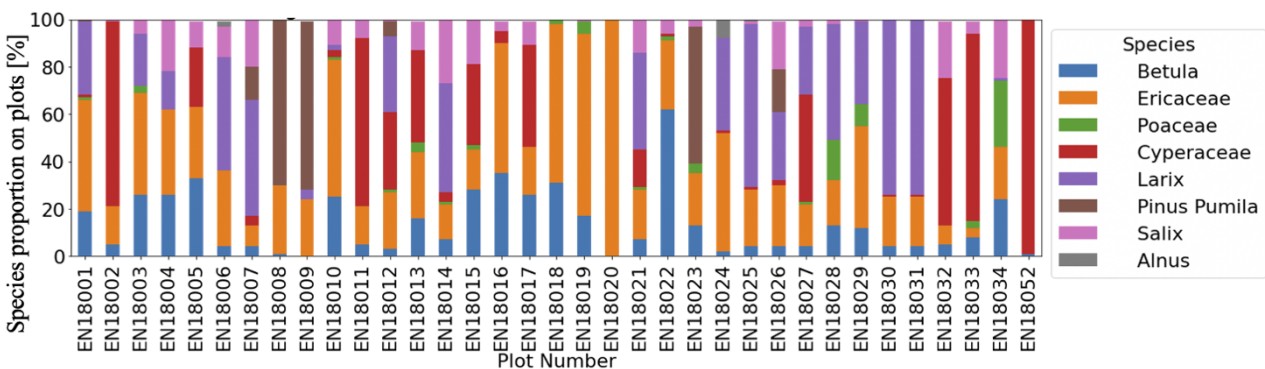


**Figure A2: Percentage vegetation cover per plot in Chukotka for all recorded vegetation in the plots.**



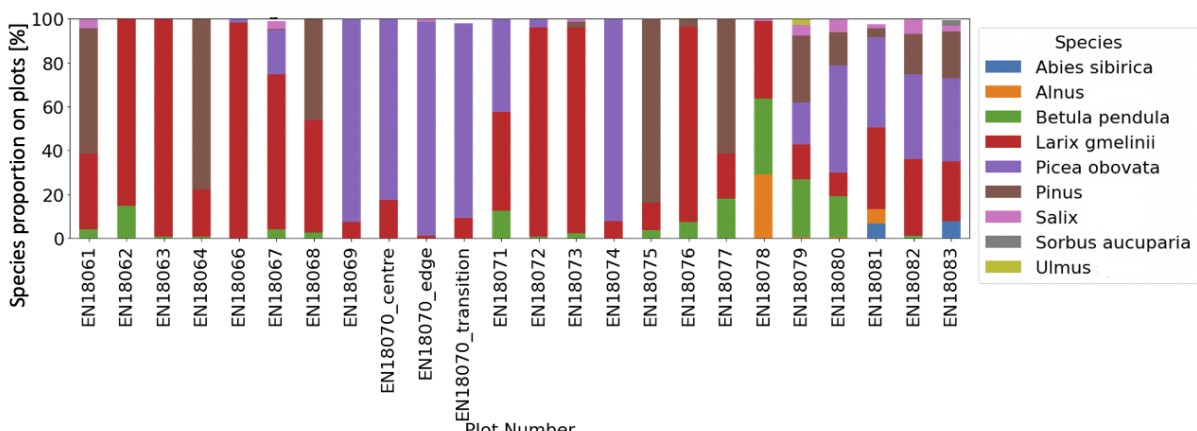

**Figure A3: Percentage vegetation cover per plot in Yakutia for only large shrubs and trees (>1.3m).**

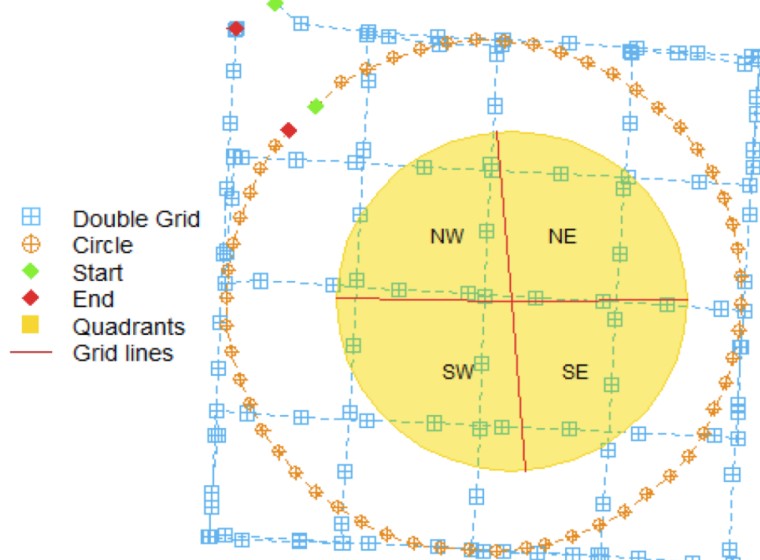


**Figure A4: SiDroForest unmanned aerial vehicle (UAV) data acquisition and flight pattern consisting of a double grid (blue) and a circular mission (orange). The two 15 m long grid lines (red) divide the plot area into four quadrants of similar size (yellow). From Brieger et al. (2019).**

"masks": {"images/00000000.jpg": {"mask": "masks/00000000.png", "color_categories": {"(255, 0, 0)": {"category": "larch", "super_category": "tree"}
{"info": {"description": "SiDroForest: Synthetic Tree Crowns", "url": "http://immersivelimit.com/datasets/test", "version": "1", "year": 2021, "contributor": "Femke van Geffen", "date_created": "12/04/2021"}"00000000.jpg", "width": 448, "height": 448, "id": 0}

**Figure A5: Example of common objects in context (COCO) style annotation labels for the masks (top) and images (bottom).**

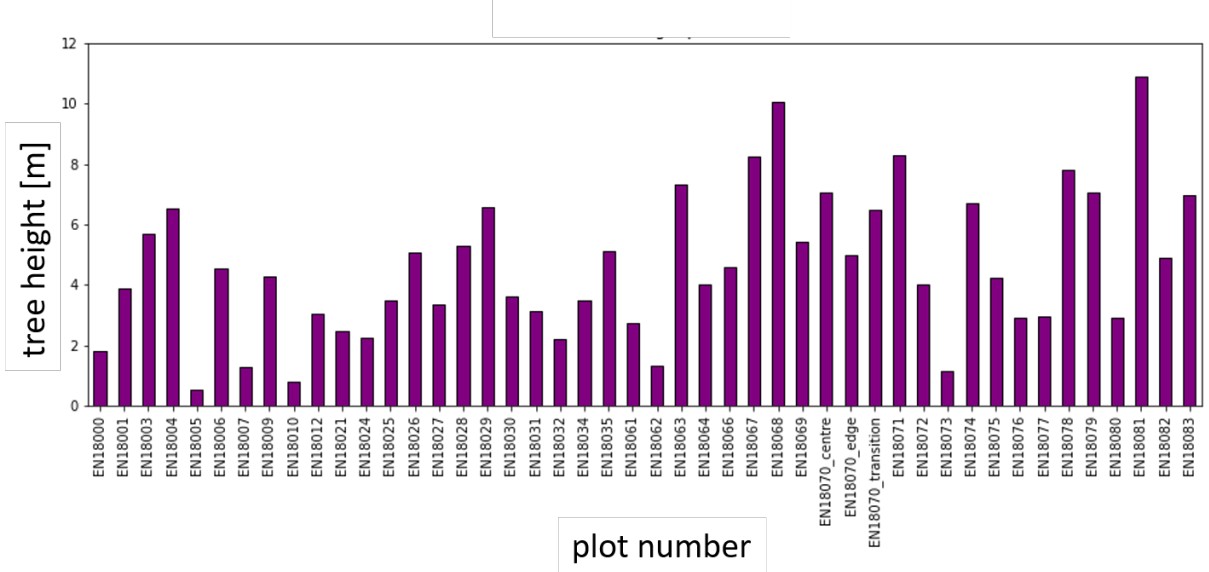

**Figure A6: Mean tree height (m) per plot from fieldwork measurements**

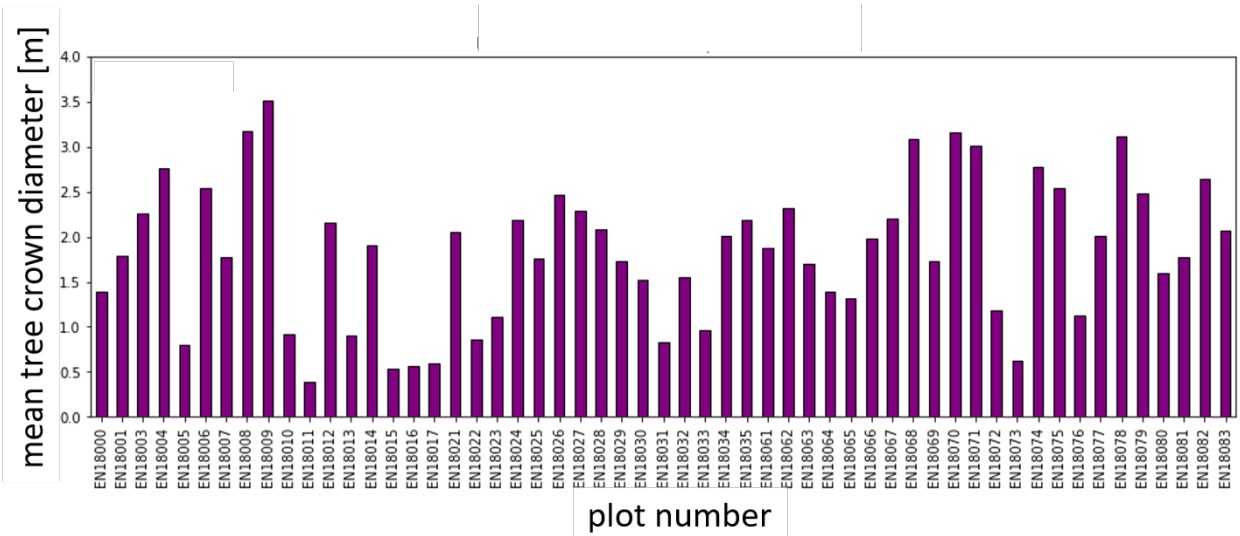

**Figure A7: Mean tree crown diameter (m) per plot from fieldwork measurements.**



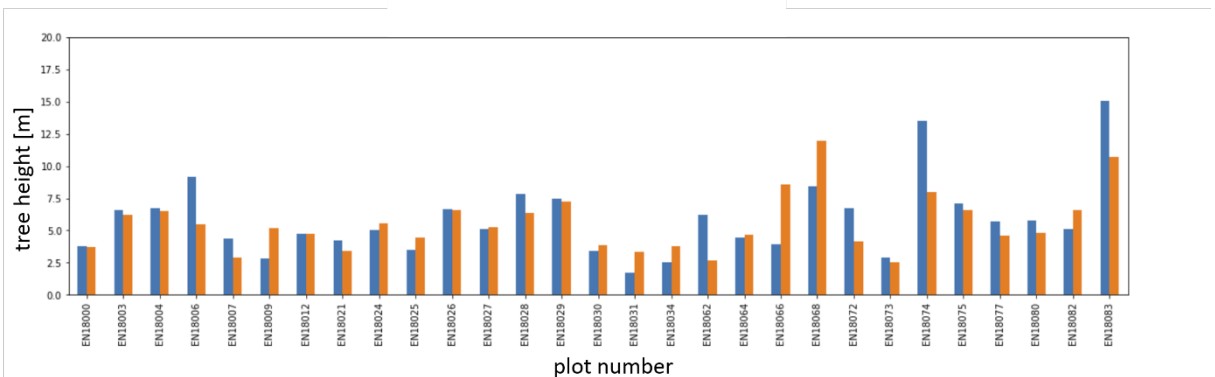

**Figure A8: Mean heights for trees and shrubs below 1.3 m for unmanned aerial vehicle (UAV)-derived heights (blue) and fieldwork-derived heights (orange).**

**Table A1: Overview of Sentinel-2 spectral bands, spatial resolution, and the central wavelength.**

| Sentinel-2 Bands | Central Wavelength (nm) | Resolution (m) |
|---|---|---|
| Band 1- Coastal aerosol | 443 | 60 |
| Band 2- Blue | 490 | 10 |
| Band 3- Green | 560 | 10 |
| Band 4- Red | 665 | 10 |
| Band 5- Vegetation Red Edge | 705 | 20 |
| Band 6- Vegetation Red Edge | 740 | 20 |
| Band 7- Vegetation Red Edge | 783 | 20 |
| Band 8- NIR | 842 | 10 |
| Band 8A- Vegetation Red Edge | 865 | 20 |
| Band 9- Water vapour | 945 | 60 |
| Band 10- SWIR-Cirrus | 1.375 | 60 |
| Band 11- SWIR-1 | 1.610 | 20 |
| Band 12- SWIR-2 | 2.190 | 20 |








**Table A2: Screenshot of the *crowns_polygon* shapefile attribute table for plot EN18077 as an example. Height: tree height in metres as identified with the tree top finding algorithm, crownAr: area of the tree crown in square metres, CrwnDmt: simplification of the crown diameter in metres assuming a circular crown, orgHght: maximum height value in metres recorded in the canopy height model (CHM) under the total crown polygon.**

| | layer | height | winRads | crownAr | crwnDmt | orgHght |
|---|---|---|---|---|---|---|
| 1 | 1.000000000000... | 3.9404767089411... | 0.97732145190239 | 2.1708000000041... | 1.6625126777775... | 4.649345874786... |
| 2 | 1.000000000000... | 0.476775185929... | 0.821454883366... | 0.388799999697... | 0.703587616866... | 0.78668737411499 |
| 3 | 1.000000000000... | 0.615317266848... | 0.827689277008... | 0.194400000005... | 0.497511575246... | 1.480337023735... |
| 4 | 1.000000000000... | 4.726099067264... | 1.012674458026... | 2.624400000362... | 1.827974250820... | 5.248609066009... |
| 5 | 1.000000000000... | 1.785895599259... | 0.880365301966... | 0.323999999974... | 0.642284681790... | 3.317377567291... |
| 6 | 1.000000000000... | 2.465017358462... | 0.910925781130... | 0.939600000241... | 1.093771400494... | 3.295063734054... |

**Table A3: An overview of the plots, the latitude and longitude of the central co-ordinate, the plot (closest lake or city), the region (Chukotka or Yakutia), and the vegetation class labels (for 30 × 30 m patches).**

| Plot Code | Latitude | Longitude | Site | Region | Vegetation Class |
|---|---|---|---|---|---|
| EN18001 | 67.39273 | 168.34662 | Lake Ilerrny | Chukotka | 2 |
| EN18002 | 67.38677 | 168.33673 | Lake Ilirney | Chukotka | 1 |
| EN18003 | 67.39273 | 168.34702 | Lake Ilirney | Chukotka | 2 |
| EN18004 | 67.39748 | 168.35122 | Lake Ilirney | Chukotka | 2 |
| EN18005 | 67.41965 | 168.3875 | Lake Ilirney | Chukotka | 1 |
| EN18007 | 67.40327 | 168.37196 | Lake Ilirney | Chukotka | 1 |
| EN18008 | 67.40213 | 168.37528 | Lake Ilirney | Chukotka | 2 |
| EN18009 | 67.40072 | 168.37968 | Lake Ilirney | Chukotka | 2 |
| EN18011 | 67.40404 | 168.36425 | Lake Ilirney | Chukotka | 1 |
| EN18012 | 67.40214 | 168.37807 | Lake Ilirney | Chukotka | 2 |
| EN18013 | 67.40517 | 168.35530 | Lake Ilirney | Chukotka | 1 |
| EN18014 | 67.39530 | 168.34910 | Lake Ilirney | Chukotka | 2 |
| EN18015 | 67.42037 | 168.33061 | Lake Ilirney | Chukotka | 1 |
| EN18016 | 67.42672 | 168.39004 | Lake Ilirney | Chukotka | 1 |
| EN18017 | 67.43229 | 168.38337 | Lake Ilirney | Chukotka | 3 |
| EN18018 | 67.456295 | 168.405961 | Lake Ilirney | Chukotka | 2 |
| EN18019 | 67.457073 | 168.408963 | Lake Ilirney | Chukotka | 1 |
| EN18020 | 67.459159 | 168.411934 | Lake Ilirney | Chukotka | 2 |
| EN18021 | 67.392129 | 168.32881 | Lake Ilirney | Chukotka | 1 |
| EN18022 | 67.401024 | 168.34800 | Lake Ilirney | Chukotka | 2 |
| EN18023 | 67.399236 | 168.35128 | Lake Ilirney | Chukotka | 1 |
| EN18024 | 67.370964 | 168.42636 | Lake Ilirney | Chukotka | 2 |
| EN18025 | 67.367027 | 168.42381 | Lake Ilirney | Chukotka | 2 |
| EN18026 | 67.396089 | 168.35429 | Lake Ilirney | Chukotka | 2 |
| EN18027 | 67.393408 | 168.35905 | Lake Ilirney | Chukotka | 2 |





| Plot Code | Latitude | Longitude | Site | Region | Vegetation Class |
|---|---|---|---|---|---|
| EN18028 | 68.467811 | 163.35762 | Bilibino | Chukotka | 1 |
| EN18029 | 68.465606 | 163.35226 | Bilibino | Chukotka | 1 |
| EN18030 | 68.405539 | 164.53273 | Bilibino | Chukotka | 2 |
| EN18031 | 68.404918 | 164.54535 | Bilibino | Chukotka | 1 |
| EN18032 | 68.404868 | 164.55118 | Bilibino | Chukotka | 2 |
| EN18033 | 68.403212 | 164.55180 | Bilibino | Chukotka | 2 |
| EN18034 | 68.403486 | 164.54804 | Bilibino | Chukotka | 1 |
| EN18035 | 68.403166 | 164.59093 | Bilibino | Chukotka | 2 |
| EN18051 | 67.802610 | 168.70471 | Lake Rauchuagytgyn | Chukotka | 1 |
| EN18052 | 67.799410 | 168.7083 | Lake Rauchuagytgyn | Chukotka | 1 |
| EN18053 | 67.797290 | 168.7107 | Lake Rauchuagytgyn | Chukotka | 1 |
| EN18054 | 67.797660 | 168.6904 | Lake Rauchuagytgyn | Chukotka | 1 |
| EN18055 | 67.791030 | 168.682500 | Lake Rauchuagytgyn | Chukotka | 3 |

| Plot Code | Latitude | Longitude | Site | Region | Vegetation Class |
|---|---|---|---|---|---|
| EN18061 | 62.076376 | 129.618586 | Yakutsk | Central Yakutia | 6 |
| EN18062 | 62.179065 | 127.805796 | Magaras | Central Yakutia | 10 |
| EN18063 | 63.776636 | 122.501003 | Vilnuyi | Central Yakutia | 10 |
| EN18064 | 63.814594 | 122.209683 | Vilnuyi | Central Yakutia | 4 |
| EN18065 | 63.795223 | 122.443715 | Vilnuyi | Central Yakutia | 9 |
| EN18066 | 63.7971189 | 122.438071 | Vilnuyi | Central Yakutia | 9 |
| EN18067 | 63.076368 | 117.975342 | Nyurba | Central Yakutia | 8 |
| EN18068 | 63.074232 | 117.98207 | Nyurba | Central Yakutia | 7 |
| EN18069 | 63.173288 | 118.132507 | Nyurba | Central Yakutia | 11 |
| EN18070 | 63.082914 | 117.984905 | Nyurba | Central Yakutia | 11 |
| EN18071 | 62.225093 | 116.275603 | Suntar West | Central Yakutia | 8 |
| EN18072 | 62.199571 | 117.379125 | Suntar | Central Yakutia | 10 |
| EN18073 | 62.188712 | 117.409917 | Suntar | Central Yakutia | 9 |
| EN18074 | 62.215192 | 117.021599 | Suntar | Central Yakutia | 11 |
| EN18075 | 62.696991 | 113.676535 | Mirny | Central Yakutia | 7 |
| EN18076 | 62.70089 | 113.67341 | Mirny | Central Yakutia | 10 |
| EN18077 | 61.892568 | 114.288623 | Mirny-Lensk | Central Yakutia | 5 |
| EN18078 | 61.575058 | 114.29995 | Mirny-Lensk | Central Yakutia | 10 |
| EN18079 | 59.974919 | 112.958985 | Lake Khamra | Central Yakutia | 8 |
| EN18080 | 59.977106 | 112.961379 | Lake Khamra | Central Yakutia | 7 |
| EN18081 | 59.970583 | 112.987096 | Lake Khamra | Central Yakutia | 8 |
| EN18082 | 59.97764 | 112.98218 | Lake Khamra | Central Yakutia | 7 |
| EN18083 | 59.974714 | 113.002874 | Lake Khamra | Central Yakutia | 7 |

1 = Graminoid tundra; 2= Forest tundra and shrub tundra; 3= Prostrate herb tundra; 4= Open canopy pine with lichen; 5= Open canopy pine;6= Closed canopy pine; 7= Open canopy mixed forest; 8= Closed canopy mixed forest; 9 = Open canopy Larch; 10= Closed canopy Larch; 11= Closed canopy spruce

745




750

## 8.Author contributions

Femke van Geffen FvG is the leading author of this manuscript and of most of the related data publications in the PANGAEA data repository. FvG wrote the manuscript together with Stefan Kruse SK, Birgit Heim BH, and Ulrike Herzschuh UH. Luidmila A Pestryakova LP and Evgenij S Zakharov EZ organised and facilitated the data collection for the expedition in Siberia and took part in the data collection. The majority of vegetation related ground fieldwork was performed by Iuliia A. Shevtsova IS, Luise Schulte LS, Simone Stünzi SS, Elena I. Troeva ET, Nadine Bernhardt NB, UH, and SK.

SK and Frederik Brieger FB undertook the data processing and together with assistants constructed the products for the orthomosaics dataset, including the point-cloud products. Rongwei Geng RG and FvG supplemented the orthomosaics dataset and assigned vegetation labels to the plots based on vegetation classes by IS. BH and Bringfried Pflug BP processed the Sentinel-2 dataset. FvG created the synthetics dataset and identified the individuals in the individual labelled trees dataset. FvG cleaned, compiled, and constructed all four final datasets under supervision of SK as lead scientist on this project.

## 9. Competing Interests

The authors declare that they have no conflict of interest.

## 10. Acknowledgements

SiDroForest was created as part of a PhD project within the context of the HEIBRiDS graduate school. This research and particularly the field work has received funding from the ERC consolidator grant Glacial Legacy of Ulrike Herzschuh (grant no. 772852).

We thank our Russian and German colleagues from the joint Russian–German expedition 2018 for support in the field. Special thanks to the staff of the BIOM-laboratory in Yakutsk for their great overall support and scientific contributions. We thank Guido Grosse and Thomas Laepple (AWI) who provided us with computational resources for the point-cloud reconstruction from UAV-based data.



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
