# Peer review of "SiDroForest: A comprehensive forest inventory of Siberian boreal forest investigations including drone-based point clouds, individually labelled trees, synthetically generated tree crowns and Sentinel-2 labelled image patches"

_Earth System Science Data, 2021_

## Author Response (AR1)

**SiDroForest ESSD review responses and changes**

Dear editor,
This document contains the three anonymous reviews that were published on ESSD in response to the SiDroForest manuscript. Each review is included in regular text and the changes that are included in the manuscript and the responses are written below each point in *bold and cursive* starting with the word *response.*

I, Femke van Geffen would like to, also on behalf of my team, thank all three reviewers for their time and detailed comments on the manuscript and dataset. The comments greatly improve the work and are well appreciated.

Kind regards,
Femke van Geffen

**Reviewer One:**

1. General comments:

These look like valuable datasets to make publicly available, from a vast region where there is a real paucity of data by any measure. These data applied to machine learning and other automated image analyses and interpretations should have value for remote sensing studies in the circumboreal region. I encourage publication, and could only note a few minor issues detailed below.

1. Significance
    1. Uniqueness

These data are definitely unique in that they are from Siberia, a larger expanse of tundra–taiga than in N. America, and yet almost all available ABOVE datasets are based in N. America, especially Alaska.

1. Usefulness.

Is it useful or necessary to display images that were excluded from the dataset?

***Response: This is included for informing on the quality of the data sets and to show details on our data processing standards: It shows the steps that were taken to make the dataset useful. Taking the images out that include people eliminates noise from the data such as the image shown in Fig. 8.***

*We also added the sentence:*

*L317-319:"The first fifteen images were excluded because they often contain a visual representation of the research team (for example, Fig. 8). Excluding these images reduces noise in the dataset as we aimed to include only forest and natural terrain in the images."*

- Completeness.

Figure 1. The study region names Chukotka and Yakutia are missing from the map or caption.

***Response: Thank you for pointing this out, the regions are now included in the figure caption below figure 1.***

Table 1. If my math is correct, it looks like Lake Ilirney should have 27 plots not n=25 as stated.

***Response: This is correct there are 27 plots. The mistake was made in the Bilibino plots where plot EN18033 and EN18035 are actually not included due to lack of usable data for these plots. That led to a mistake in the total which led to a mistake for the plots of Lake Ilirney, which in fact does have 27 plots. It is now corrected in Table 1.***

1. Data quality

L243. Were the cardinal directions adjusted for magnetic declination in the field, or prior to preparing the data product? This will affect how the plot sample area intersects with any georectified remotely sensed data.

***Response: The magnetic declination in central Yakutia and Chukotka is very low (around or less than 10 °). We oriented the vegetation plot by iteratively moving with GPS with the set up for the true North, not the magnetic North, and did not use a compass for the North orientation of the vegetation plot. Like this, despite also the challenge of accurate GPS measurements in the field, we consider the North-orientation of the vegetation plot as robust and reliable***

Presentation quality

Fig. 6. The background RGB image is shifted between the images on the left and right, making it unnecessarily difficult to visually compare the overlaid features and labels.

***Response: Yes, thank you, The figure was redone (please see below). Two figures were combined into one new* Fig. 6*, hopefully a clearer image.***

[Figure]

Legend
- • Individual tree point
- ▢ Species polygon

2. Specific comments:

L244-245. Could tree density per plot be calculated from these data? It sounds like the answer is no, if the only requirement was to sample 10 trees. How were the 10 trees selected from the size and spatial distribution of available trees in the plot?

*Response: Yes, tree density per plot can be calculated from the data. The 10 trees were only selected to cover the whole size range of each species and investigate in-depth. All trees on the plots, with a height of >40 cm, were recorded and their height, vitality status, and species noted. This information per plot can be easily used to calculate the density.*

*In addition, the point clouds contain all trees on the plots. This information can also be extracted.*

L556. Were the estimations made in the field or in the office based on visual interpretation of imagery?

*Response: Thank you for your feedback. We have clarified it in the text.*

*The recorded tree crowns refers to the trees measured in the field that were then linked manually to the orthoimages based on their locations and visual appearance. L410-415:*

*"3.2 Dataset 2: Individual labelled trees*

*In order to make assumptions and predictions about the content of the vegetation plots it is important to link the labelled individual trees from the fieldwork to the processed orthoimages. We located 872 trees and large shrubs in the orthoimages that were surveyed in Siberia during the two-month fieldwork expedition in 2018 (Kruse et al., 2019) (Fig. 16). "*

**The automatically detected tree crowns were estimated with detection software in R on the UAV orthoimages. Explained in the text at L273-279:**

*" The SiDroForest data collection also contains 19 342 automatically detected tree-crown polygons (Kruse et al. 2021b). The tree crowns were captured in the CHM by watershed segmentation analysis using the R package ForestTools (Plowright, 2019) and successive automatic generation of a polygon around them following Brieger et al. (2019). This automated tree-crown detection algorithm was run for all plots and the resulting shapefiles are provided for each plot that contained trees. Quality assurance was performed for each plot by carefully examining each plot based on expert knowledge and assigning a quality score of Q1 (good quality), Q2 (medium quality), or Q3 (poor quality) to the shapefile products."*

3. Technical corrections:

L82. Not just a C sink, but also potentially a C source.

L93-94. This sentence is incomplete.

L103. The word "Likeness" actually isn't and should be deleted.

L128. Change "was" to "were".

L129. Change "was" to "were".

L153. The sentence that ends here needs a period.

L203-204. State why datasets one and two were prepared before doing this for datasets three and four.

L240. Change "projective" to "projected".

L253. Change "are" to "is".

L618. Should not the citation be Fig. 22 (instead of Fig. 23)?

L672. Change "as" to "a".

**Response: All changes were made based on the technical corrections suggested.**

**Reviewer Two:**

The datasets presented seem to be quite useful for an important region where data is scarce for research. I believe these field survey data would be helpful for both small-scale and large-scale studies in understanding the circumboreal region. A few comments listed below for consideration.

Q1: Is the data set significant – unique, useful, and complete?

I think the data is unique and complete, though the authors can mention more about the usefulness of these datasets. For example, some scientific questions that these datasets can help address? What specific applications can be expected from these datasets? Or how these datasets may help change the research landscape over time?

***Response: We highlight the usefulness of the data set with first information in the introduction and describe its uniqueness and suggestions to how to use the data in the discussion. The purpose of each data type is also mentioned in the product data sections. I have clarified the usefulness more in the current version of the paper, please see lines below:***

***1-The satellite data is processed to a high degree and prepared for machine learning tasks with labeled patches.***

***L56:*** *"The fourth dataset contains Sentinel-2 Level-2 bottom of atmosphere processed labelled image patches with seasonal information and annotated vegetation categories covering the vegetation plots (van Geffen et al., 2021b, https://doi.pangaea.de/10.1594/PANGAEA.933268). The dataset is created with the aim of providing a small ready-to use validation and training data set to be used in various vegetation-related machine-learning tasks. It enhances the data collection as it allows classification of a larger area with the provided vegetation classes. "*

***2- The orthoimages provide geographical information on individual trees and shrubs that were recorded on each plot. These individuals have information such as tree height and species and can be used to calculate the projected cover of each species and give general insights into the vegetation dynamics.***

***L153:*** *"Individual labelled trees surveyed during the fieldwork, including information on height, tree crown, and species. These tree-individual labelled point and polygon shape files (light green symbols) were generated and are linked to the UAV RGB orthoimages of the expedition vegetation plots."*

*3- The point clouds provide information on the 3D structure of the forest at each location. The point cloud products such as Canopy Height Model can be used to extract the height of all the trees on the plots, not just the recorded ones from fieldwork.*

*L36-41 "The first dataset provides Unmanned Arial Vehicle (UAV)-borne data products covering the vegetation plots surveyed during fieldwork (Kruse et al., 2021, https://doi.pangaea.de/10.1594/PANGAEA.933263). The dataset includes structure from motion (SfM) point clouds and Red Green Blue (RGB) and Red Green Near Infrared (RGN) orthomosaics. From the orthomosaics, point-cloud products were created such as the Digital Elevation Model (DEM), Canopy Height Model (CHM), Digital Surface Model (DSM) and the Digital Terrain Model (DTM). The point cloud products provide information on the three-dimensional (3D) structure of the forest at each plot. "*

*4- The synthetic dataset provides 10000 images that were generated using an algorithm. The purpose for this is that they can now be used as input for a neural network which needs many images as input for training.*

*L51-55: "The third dataset contains a synthesis of 10 000 generated images and masks that have the tree crowns of two species of larch (Larix gmelinii and Larix cajanderi) automatically extracted from the RGB UAV images in the common objects in context (COCO) format (van Geffen et al., 2021a, https://doi.pangaea.de/10.1594/PANGAEA.932795). As machine learning algorithms need a large dataset to train on, the synthetic dataset was specifically created to be used for machine learning algorithms to detect Siberian larch species."*

Q2: Is the data set itself of high quality?

Yes, the data is of high quality.

Presentation quality

I do not think the authors pointed out specific/suitable software for simple visualization and analysis as I typically need to install some software to view the data.

*Response: Yes, we did not specifically named suitable software. The Sentinel-2 and UAV orthoimages data can also be opened and viewed in any open source and commercial GIS and Remote Sensing software and for the point clouds AgiSoft or Cloudcompare can be used. All data can be loaded in R. Suggestions to software have been added to chapter 2 data and methods and to specific sections*

L165-170: " *The SiDroForest products are in common software formats: there are point and polygonal shape files (shp), raster files are in the georeferenced tagged image format (tif), Geotiff, shapefile formats and*

*JavaScript Object Notation (JSON) can be read and visualized in any open source and commercial GIS and Remote Sensing software tools and a wide range of libraries in R, python and other programming languages. The point clouds are provided in the standard LASer (LAS) binary file format that can be handled in any software that supports this format such as CloudCompare (CloudCompare, 2021) or R (R, 2020) or Python libraries specifically developed for this datatype."*

Specific comments

(1) The abstract can be improved by why these datasets are needed.

**Response: We enhanced the abstract text in making our statements clearer. Limited data on boreal forest structure are available, especially for this region (Eastern Siberia) and in quality with labels that can be used for machine learning purposes. Please find all the changes in the revised manuscript with tracked changes.**

(2) The abstract seems too long and I would suggest further summarizing it and highlighting the datasets in brief.

**Response: We have taken this suggestion into consideration. Please see new document with the shortened, edited abstract.**

[revised manuscript text omitted]

(3) I am not convinced how this field level data can help studies in this region. Only for optimized data containing annotated vegetation categories. What is the motivation for doing these machine learning applications? DO we need to look at data from a temporal perspective?

*Response: The idea is that more detailed vegetation information can be extracted from this dataset per plot. Also, there is information on the vegetation structure of each plot in the SfM point clouds.*

*The machine learning applications for tree crowns created a set with n = 10 0000 crowns that can be used to automatically detect larches in UAV RGB images of forest plots.*

*It depends on the application if we should include temporal perspectives. The temporal information in the current dataset is the three seasons of satellite data included in the Sentinel-2 part of the dataset. At the current state this data collection is now an important and unique openly published data collection, a snapshot for the time window around 2018 for the taiga-tundra transition zone in Chukotka and the evergreen-summergreen transition zone in Central Yakutia. if in the future again plots of the same region will be assessed, change can be detected. This time stamp in 2018 can also support the assessment of land surface satellite products of the second late decade of this century compared to later satellite acquisitions in future and earlier satellite acquisitions in the past.*

(4) The introduction should be shortened as well to highlight the contribution of datasets to the scientific literature or specific questions/challenges to be addressed

*Response: Thank you for your feedback. We have rewritten the introduction and shortened it considerably (please see the track change file).*

(5) Figure 1 suggested sampled sites are very limited. Can the authors justify why these sites were selected rather than other sites?

*Response: The sites were selected based on the two important transition zones in these areas. The **summergreen-evergreen and the Tundra-Taiga transition zone. The sites included in this dataset are from one very extensive, 2 month long expedition in 2018 where the team traveled from the City of Yakutsk following the only possible summer road in the direction of Mirny and then Lensk. We then followed the bioclimatic gradient leading to the easternmost larch dominated forests and westernmost mixed-species forests in the Lake Khamra region, this expedition part that assessed xxx forest plots took 4 weeks and was very efficient. From the Tundra-Taiga transition zone there is only summer road close to River Kolyma to Bilibino and then the expedition team needed to take helicopters to the mountainous tundra and forest tundra regions of Lake Rauchagytgyn and Lake Ilirney. This part of the expedition also took 4 weeks and was very time intensive with a lot of plots assessed (n= x). The sites were selected beforehand based on satellite imagery using the NDVI maximum summer values and change as well as disturbance products like Hansen et al. forest loss/gain for selecting various accessible sites from the road. This information has now also been added to the paper to make it more clear why we selected these sites. The 2018 two month-long expedition was a very extensive expedition covering a variety of environments with an unusual high number of field plots with successful UAV acquisitions.***

(6) I am not sure whether the codes for generating these datasets are available as it is not mentioned explicitly in the manuscript.

*Response: The code to generate the Synthetic tree crowns is referred to in the text, Kelley (2019) and the link to the github with the code is in the reference list: GitHub repository: https://github. com/akTwelve/cocosynth, 2019. **This code can be downloaded and used to make the synthetic dataset. All code is in Python.***

*The individual trees and the Sentinel-2 were generated by hand and with free remote sensing and GIS (QGIS) software now put in in chapter 2. The SfM point clouds were made with Agisoft software, described in chapter 2. The tree crown detection was implemented in R which we describe and refer to Brieger et al. (2019) where it is described in much detail.*

**Reviewer Three:**

General comments

The author collected tree - and plot-level forest structure data based on unmanned aerial vehicle and field investigation in two vegetation transition zones of Siberia, Russia. The datasets, including field plot level individual tree and shrub records (tree height, crown diameter, and species) and UAV products (e.g., Canopy Height) can be used for calibration and verification of model output, experiments, or observations. They are useful and important for future carbon dynamic studies and help to inform forest management, especially for the area where historical records and monitoring tend to be scarce. However, the manuscript is poorly organized and difficult to read. I encourage publication after addressing the following issues.

Significance:

The data are useful, complete, and fill in the region field data gap for Siberia boreal forests.

Data quality:

The data are easy to understand and presented readily and accessible to be used in other studies.

Presentation quality:

The manuscript was not logically articulated and was poorly written.

I recommend reframing the Introduction section. It should be more concise. I suggest shortening the detailed SiDroForest dataset and collection method description, but introducing the necessary or implication of each dataset (such as tree-level or plot–level records or canopy Height) and discussing the importance and challenges to collect these data.

***Response: We have taken this suggestion into consideration. Please see new structured manuscript with the tracked changes that has been submitted.***

The results seem like a duplicate of the method. I suggest including some further information and analysis, not only what the data were included or collected in the four datasets. For example, the frequency or distribution of tree species in the area for 3.2 Dataset 2, and the dominant vegetation classes and their distribution for Dataset 4 can be described.

***Response: Thank you for your feedback. In fact, we detected some repetitions and optimized the data and methods and result section. We removed repetitive***

*information and put highlights into the result chapter. Several of the results are visualized in the Appendix, e.g. the distribution of vegetation species in A2, and Percentage vegetation cover per plot in Yakutia for only large shrubs and trees (>1.3m) in A3. The dominant vegetation class of the Sentinel-2 patches are shown in Table 3. Please also see the track change file as an overview on our edits in chapter 2 'data and methods and chapter 3 'results'.*

Specific comments:

Line 68: change "are" to "is"

Lines 106-109: The sentence is difficult to read.

Lines 119-121: delete one of the "use" in the sentence

Line 124: add "which was" before "derived from"

Line 129: change "was" to "were"

Line 150: add"," before "and"

*Response: Above specific comments were all taken into account and corrected for.*

Lines 243-244 and Figure A4: Two 30-m-long tape or 15m?

*Response: We investigated a circular plot area and used two 30-m long tape measures to segment the circle with a 15 m radius into four quadrants.*

*L195-200* *"In the field, two 30-m-long tape measures were laid out along the main cardinal directions, intersecting in the plot centre, marking the main axes of a circular area with a radius of 15 m. A minimum of ten individuals of each tree and shrub species present were selected per plot. For each individual tree we measured the stem diameter at breast height and at the base. The tree crown diameter, tree height, and vitality were estimated as described in Brieger et al. (2019). There were three deviations from the standard method of vegetation inventory. On plot EN1814 and EN1865, all trees were recorded, and plot EN18070 was recorded by a transect with three segments: edge, transition, and centre."*

Lines 243-248: why were a minimum of ten individuals selected per plot? Why were plots recorded differently? If only part of the trees and shrubs were recorded, the data may not be able to represent the real forest information.

*Response: All plots were recorded with the same method. All trees present on the plot area were recorded in the field by its height, species and vitality information. Additionally, a minimum of 10 individuals per species were selected for in-depth analyses and can be found in the Species polygons. Shrub cover per taxa was recorded in the field as well but only a minimum of three individuals selected for in-depth analyses. The UAV-derived Species polygon is supplied for classification tasks*

*and marks a clear tree or shrub from a certain species. It includes a minimum of 10 recorded trees or shrubs per species, where possible the same as recorded in fieldwork.*

Line 252 please clarify the 11 vegetation classes here.

*Response:*

*A clarifying sentence is added to L201-204: "We post-fieldwork assigned 11 vegetation classes to the 64 plots (table A1). The class assignment was based on the previous classes determined by Shevtsova et al. (2020a) for Chukotka. For plots in Central Yakutia, we applied a similar method incorporating principal component analysis (PCA), tree density information from the UAV data, and recorded tree species information per plot (Fig. A2, A3 show the field data information)."*

Lines 256-259 and Figure A4: The vegetation plot looks smaller than 30m ×30m (the double grid) if the red line is 15m long.

[Figure]

*Response: Yes, the double grid in figure A4 is not the 30 m x 30 m vegetation plot, but the UAV double grid flight line set up around the 15 m radius vegetation plot, the blue grid covers around 50x50 m square overflight grid and the orange circular flight mission with a radius of ca. 25 m around the center was carried out with the camera pointing towards the center.*

*Figure A4: SiDroForest unmanned aerial vehicle (UAV) data acquisition and flight pattern consisting of a double grid (blue) and a circular mission (orange) around the vegetation plot.*

[Figure]

*Figure A1 The two 15 m long grid lines (red) divide the plot area into four quadrants of similar size (yellow). From Brieger et al. (2019).*

Line 274: with very low vegetation?

*Response: The vegetation that is near to the ground i.e., low structure vegetation. Often this is small shrubs and small trees that are hard to segment because they are close together. We changed the text accordingly to 'low structure vegetation'*

Line 295: add "that was" between "area" and "not"

Lines 301-304: Please break the long sentence into several simple sentences.

Line 326: add "that were" before "corrected"

*Response: All three suggestions above were followed.*

Lines 338 -340: the sentence is difficult to read. Does that mean the tree crowns were captured by two methods: 1) watershed segmentation analysis and 2) successive automatic generation of a polygon around them?

*Response: Yes. I corrected the sentence to be clearer.*

Line 542: Link or Linking?

*Response: Link.*

Lines 553-557: what's the meaning of this paragraph? Do you compare your field-measured crown diameters and detected crown polygons? If so, what's the difference between your results and the results of Brieger et al.?

**Response: The paragraph introduces that the automatically detected tree crowns are likely better fitting than the field estimations for each tree. We present here the detected tree crowns and did not run an analysis such as in Brieger et al. 2019.**

---

## Referee Report (RR1)

This manuscript presents an UAV-based dataset collection of forest structure over two vegetation transition zones of Siberia, Russia. This dataset can support forest carbon dynamic studies and machine learning based land cover mapping. However, the manuscript is poorly written and read like a technical report.

**Is the data set significant – unique, useful, and complete?**

I think this dataset is definitely unique given the data scarcity over boreal forest region. However, this data set misses a very important information about acquisition date as this is quite important to match satellite images to avoid phenology impacts.

**Is the data set itself of high quality? Is the data set publication, as submitted, of high quality?**

I think the data set itself is not as good as the one presented in the manuscript.

I downloaded the datasets and checked some plots, and could not reproduce the same results. For example, the CHM of plots EN18077 shows CHM range and spatial pattern very different from Figure 13 in the manuscript.

[Figure]

**Specific comments:**

L82. Forest could also be a C source especially has been disturbed recently.

L224, it will be great to provide the specific measurement dates for each plot or least given date range for these two transition zones. Time information sometimes is equally important as geographical coordinates.

L245, what rules were used to select the minimum of 10 individuals. considering the largest 10 ones?

L246, EN1814 and EN1865 are not found in Table 1. Are they EN18014 and 18065?

L303, "... using a Cloth Simulation Filter (Zhang et al., 2016) …" has been explained in L273. Please describe the generation of groudonly and treeonly processes only once. Same thing to "Agisoft PhotoScan Professional" in L289 and L267.

L310, 'in R..' is also mentioned in L290. Please remove the repeated information.

L323-330, were photons of all plots taken the same day? If not, was there any corrections to photons to match color histogram between plots? Please make this clear.

L348, why mentioned ".. during the two-month fieldwork expedition in 2018 (Kruse et al., 2019).." here again since it has been introduced in L224? Were they different expeditions?

L334, I think the affected plots are not just these three. For example, RGB orthomosaics for plot EN18000 have many blurry parts over the canopy of some trees. BTW, EN1878 and EN1879 are not listed in Table 1. Missing a digit?

L343, please mention that some plots do not have even Q1 shapefiles (e.g., EN18007).

L460, what is the unit of RGM_CHM file? I checked the EN18077_RGB_CHM.tif file and the pixel value ranges from 0.0108667 to 0.637536, which is quite different from Figure 13.

---

## Author Response (AR2)

**SiDroForest ESSD review responses and changes**

Dear Editor,

This document contains the answers to the editorial request (minor revisions) related to the anonymous reviewer's comments (reviewer N°4) in response to the SiDroForest manuscript. Each reviewer's statement is included in regular text and the responses are written below each point in *cursive* .

I, Femke van Geffen would like to, also on behalf of my team, thank the reviewer very much for the careful examination of the data. The reviewer found by this that one of the subsets of the bigger data sets was erroneous. We downloaded and checked all other published data sets - they are correct. We corrected the erroneous subset files and published the corrected files as version 2. It seems that a large share of the reviewer's ranking and comments (e.g. related to 'carbon source', or to the repetition of technical details, and the usefulness of the dataset) refer to the first version of the discussion manuscript (that is also visible by the use of the old line numbers of the first version of the manuscript), and have already been corrected for in the latest uploaded version of the manuscript that was uploaded in the system (manuscript and tracked change manuscript). The new edits related to the reviewer's comment (N°4) make the manuscript and dataset (including the updated read me files in the data publications) further clearer and still more user friendly. The reviewer's comments improved the work and are well appreciated. Please find point by point responses below.

Kind regards,

Femke van Geffen

*Dear Reviewer*

*Response to Figure 13 comments:*
*In fact one of the subsets of the UAV product data collection Kruse et al. 2021,*
*https://doi.pangaea.de/10.1594/PANGAEA.933263  missed the elevation information per pixel in the geotiff files. This refers to the subset of the canopy height model CHM files, we had uploaded the wrong process level of the CHM files and apologize sincerely for this. We have carefully checked every file of the published products - the other data subsets are correct. We have uploaded a corrected CHM version 2 that are now linked via the original dataset:*
*https://doi.pangaea.de/10.1594/PANGAEA.933263*

*The CHM files now are the right files and are the same as the images shown in the paper. Thank you so much for pointing it out.*

[Figure]

Response to the specific comments:

L82. Forest could also be a C source especially has been disturbed recently.

*Response: Yes, this statement was already corrected in the previous iteration of the reviews: L74 in the latest version of manuscript: "Forest structure is a crucial component in the assessment of whether a forest is likely **to act as a carbon sink or source** under changing climate (e.g., Schepaschenko et al., 2021)."*

L224, it will be great to provide the specific measurement dates for each plot or least given date range for these two transition zones. Time information sometimes is equally important as geographical coordinates.

*Response: Good point. In fact we realized now by your comment that we did not include a table with the acquisition dates. We now have added the acquisition date to table A1.*

L245, what rules were used to select the minimum of 10 individuals. considering the largest 10 ones?

*Response: The selection of trees was based on how representative those tree types were for this forest type so that it represents the vegetation as well as possible. To make sure that the data is evenly distributed, we included at least 10 trees per species (if there were as many of these species on the plot).*

L246, EN1814 and EN1865 are not found in Table 1. Are they EN18014 and 18065?

*Response: Yes, thank you for pointing this out. It is now corrected to EN18014 and EN18065.*

L303, "... using a Cloth Simulation Filter (Zhang et al., 2016) ..." has been explained in L273. Please describe the generation of groudonly and treeonly processes only once. Same thing to "Agisoft PhotoScan Professional" in L289 and L267.

L310, 'in R..' is also mentioned in L290. Please remove the repeated information.

L348, why mention ".. during the two-month fieldwork expedition in 2018 (Kruse et al., 2019).." here again since it has been introduced in L224? Were they different expeditions?

*Response to the comments: We assume that these repetitions mentioned (and that all show the old line numbers from the original version of the discussion paper), have been already corrected in our previous iteration of the reviews: the latest submitted manuscript underwent a lot of cleaning and these changes (like the carbon sink, in the first comment) have already been made (in the previous review round).*

L323-330, were photos of all plots taken the same day? If not, were there any corrections to photos to match color histogram between plots? Please make this clear.

*Response: No, the UAV camera acquisitions were taken on different dates during the 2-month long expedition, when visiting the vegetation plots. The dates of the field work are now added in table A1. There was no 'color matching between acquisitions' as these were acquisitions in the field under different illuminations: overcast with now shadows as best condition for spectral imaging, and sunny with strong shadow formation (of the trees) as the least favorable condition. The cameras of every acquisition were calibrated and referenced to photo panels, however this not yet a normalization such as transferring the DN data into quasi-reflectance data that would allow to have absolute color values between acquisitions.*

L334, I think the affected plots are not just these three. For example, RGB orthomosaics for plot EN18000 have many blurry parts over the canopy of some trees. BTW, EN1878 and EN1879 are not listed in Table 1. Missing a digit?

*Response:Yes, the names were missing a 0. They are now corrected. Thank you. As for the data quality, EN18030, EN18078, and EN18079 are clear examples as is also mentioned in the paper. We describe that some sites have some parts of blurred areas in trees. This is usually related to canopy movements due to wind and cannot be avoided in the acquisition process at high latitudes in the field, where there are nearly never wind free time slots. We added this sentence.*

*'Not all RGB orthomosaics have the same high quality, as varying flight or weather conditions affected the construction of the final products. The canopy moved due to wind that cannot be avoided in the acquisition process at high latitudes in the field, where there are nearly never wind free time slots.'*

L343, please mention that some plots do not have even Q1 shapefiles (e.g., EN18007).

*Response: We downloaded the data from PANGAEA and found that EN18007 has the Quality file, in this case Q2. (As you can see in the image below). If Q1 files are not present, the Q files are of the other, lower Quality categories 2 and 3. Not all plots have trees, the tree crown product and Q files could be only produced when vegetation height was adequate. We added an overview table in the read me file in the 2nd version of the PANGAEA data publication for a more user friendly overview.*

[Figure]

*Left: orthoimage without crown polygon. Right: detected tree crowns with Q2, most crowns have been detected.*

L460, what is the unit of RGM_CHM file? I checked the EN18077_RGB_CHM.tif file and the pixel value ranges from 0.0108667 to 0.637536, which is quite different from Figure 13.

*Response: Yes in this subset of the data publication incorrect CHM files had been regrettably included and are now corrected (see explanation above). The range of EN18077 is now up to 19 m canopy height and the figure 13 showing the color coded CHM in the manuscript is also updated. Thank you very much for checking the data! This was a lot of reviewer work.*

---

## Author Response (AR3)

Dear editor,

Thank you very much for your work and support during the review process. With the suggestions of the reviewers and your feedback we feel that the manuscript has improved tremendously.
The answers to the reviewer to L245 and L323-330 have now been added into the manuscript and the dataset has been thoroughly checked. Thank you very much for your time and effort.

Femke van Geffen